# Reconstructed influenza A/H3N2 infection histories reveal variation in incidence and antibody dynamics over the life course

James A. Hay [1,2]*, Huachen Zhu[3,4,5], Chao Qiang Jiang[6], Kin On Kwok[7,8,9], Ruiyin Shen[6], Adam Kucharski[10], Bingyi Yang[11], Jonathan M. Read[12], Justin Lessler[13,14,15], Derek A. T. Cummings[13], Steven Riley[2]*

1 Pandemic Sciences Institute, Nuffield Department of Medicine, University of Oxford, Oxford, United Kingdom, 2 MRC Centre for Global Infectious Disease Analysis, Imperial College London, London, United Kingdom, 3 Guangdong-Hong Kong Joint Laboratory of Emerging Infectious Diseases/MOE, Joint Laboratory for International Collaboration in Virology and Emerging Infectious Diseases, Joint Institute of Virology (Shantou University/The University of Hong Kong), Shantou University, Shantou, China, 4 State Key Laboratory of Emerging Infectious Diseases/World Health Organization Influenza Reference Laboratory, School of Public Health, Li Ka Shing Faculty of Medicine, The University of Hong Kong, Hong Kong, China, 5 5EKIH (Gewuzhikang) Pathogen Research Institute, Guangdong, China, 6 Guangzhou No.12 Hospital, Guangzhou, Guangdong, China, 7 The Jockey Club School of Public Health and Primary Care, The Chinese University of Hong Kong, Hong Kong Special Administrative Region, China, 8 Stanley Ho Centre for Emerging Infectious Diseases, The Chinese University of Hong Kong, Hong Kong Special Administrative Region, China, 9 Hong Kong Institute of Asia-Pacific Studies, The Chinese University of Hong Kong, Hong Kong Special Administrative Region, China, 10 Department of Infectious Disease Epidemiology, London School of Hygiene & Tropical Medicine, London, United Kingdom, 11 WHO Collaborating Centre for Infectious Disease Epidemiology and Control, School of Public Health, Li Ka Shing Faculty of Medicine, The University of Hong Kong, Hong Kong Special Administrative Region, China, 12 Centre for Health Informatics Computing and Statistics, Lancaster University, Lancaster, United Kingdom, 13 Department of Epidemiology, Johns Hopkins Bloomberg School of Public Health, Baltimore, Maryland, United States of America, 14 Department of Epidemiology, UNC Gillings School of Global Public Health, Chapel Hill, North Carolina, United States of America, 15 UNC Carolina Population Center, Chapel Hill, North Carolina, United States of America

* james.hay@ndm.ox.ac.uk (JAH); s.riley@imperial.ac.uk (SR)

**Data Availability Statement:** All code and data required to reproduce the analyses are available at https://doi.org/10.5281/zenodo.12795911.

## Abstract

Humans experience many influenza infections over their lives, resulting in complex and varied immunological histories. Although experimental and quantitative analyses have improved our understanding of the immunological processes defining an individual's antibody repertoire, how these within-host processes are linked to population-level influenza epidemiology in humans remains unclear. Here, we used a multilevel mathematical model to jointly infer antibody dynamics and individual-level lifetime influenza A/H3N2 infection histories for 1,130 individuals in Guangzhou, China, using 67,683 haemagglutination inhibition (HI) assay measurements against 20 A/H3N2 strains from repeat serum samples collected between 2009 and 2015. These estimated infection histories allowed us to reconstruct historical seasonal influenza patterns in humans and to investigate how influenza incidence varies over time, space, and age in this population. We estimated median annual influenza infection rates to be approximately 19% from 1968 to 2015, but with substantial variation between years; 88% of individuals were estimated to have been infected at least once during the study period (2009 to 2015), and 20% were estimated to have 3 or more infections in

**Funding:** JAH is supported by a Wellcome Trust Early Career Award (grant 225001/Z/22/Z). This study was supported by grants from the NIH R56AG048075 (DATC, JL), NIH R01AI114703 (DATC, BY), and the Wellcome Trust 00861/Z/16/Z (SR) and 200187/Z/15/Z (SR). This work was also supported by research grants from Guangdong Government HZQB-KCZYZ-2021014 and 2019B121205009 (HZ). DATC, KOK, JL, JMR, and SR acknowledge support from the National Institutes of Health Fogarty Institute (R01TW0008246 and R01AI160780). JMR acknowledges support from the Medical Research Council (MR/V038613/1; MR/Z504373/1). KOK acknowledges support from Health and Medical Research Fund (reference numbers: INF-CUHK-1, 17160302, 18170312, 22210232, CID-CUHK-A, COVID1903008), General Research Fund (reference numbers: 14112818, 24104920), Group Research Scheme of The Chinese University of Hong Kong, and Funding Allocation to Faculties by Research Committee of The Chinese University of Hong Kong. The funders played no role in the study design, data collection and analysis, decision to publish, or preparation of the manuscript.

**Competing interests:** DATC declares research funding from Merck, Sharp and Dohme and from Pfizer for research unrelated to this manuscript. All other authors have declared that no competing interests exist.

**Abbreviations:** CoV, coefficient of variation; CrI, credible interval; HI, haemagglutination inhibition; ILI, influenza-like illness; MCMC, Markov chain Monte; MDCK, Madin-Darby canine kidney; RDE, receptor-destroying enzyme; TCID50, 50% tissue culture infectious dose.

that time. We inferred decreasing infection rates with increasing age, and found that annual attack rates were highly correlated across all locations, regardless of their distance, suggesting that age has a stronger impact than fine-scale spatial effects in determining an individual's antibody profile. Finally, we reconstructed each individual's expected antibody profile over their lifetime and inferred an age-stratified relationship between probability of infection and HI titre. Our analyses show how multi-strain serological panels provide rich information on long-term epidemiological trends, within-host processes, and immunity when analysed using appropriate inference methods, and adds to our understanding of the life course epidemiology of influenza A/H3N2.

## Introduction

Patterns of influenza infections in humans are highly varied across time, space, and demography [1,2]. Recurrent epidemics occur because influenza viruses undergo an evolutionary process of antigenic drift, whereby new strains escape preexisting host immunity through the accumulation of mutations in immunodominant surface glycoproteins leading to rapid turnover of lineages, with specific strains persisting for 1 to 2 years [3,4]. Because individuals are alive at different times and locations, they are exposed to different strains and thus each individual has a distinct immunological history [5,6]. As a result, serological data suggest that humans are infected with a new A/H3N2 influenza strain approximately every 5 years, with less frequent infections, or at least less frequent detectable antibody boosts, as individuals enter middle age [7,8].

A better understanding of who, where, and when influenza infections are likely to occur would aid in public health planning, nowcasting, and forecasting [9,10]. However, it is not just antigenic variation and evolution that contributes to variation in influenza incidence, but a combination of individual and population level factors [11,12]. Birth cohorts [13–15], contact and movement patterns [16–18], climatic variation [19,20], school terms [21,22], city structure [23,24], and household structure [25,26] have all been shown to be associated with variation in influenza incidence. However, variation in surveillance quality and consistency across locations and over time makes it difficult to identify individual-level or population-specific effects over a longer time period using routine influenza-like illness (ILI) surveillance data [27,28]. These limitations may be overcome by using serological data, where unobserved past infections and vaccinations leave a signature in an individual's measurable antibody profile [29–31].

For influenza, measured antibody levels are the result of complex interactions of immunological responses from all past exposures [6,32]. Hence, accurate inferences of individual infection histories require models of antibody kinetics to determine the number and timing of past exposures to multiple influenza strains [8,13,33–35]. These models can be complicated, as immunological interactions of antigenic drift with immune memory occur through imprinting effects, whereby the set and order of strains in an individual's previous exposure history influences which epitopes are targeted and the magnitude of their antibody response to subsequent exposures [6,32]. Estimating influenza infection histories from serological data therefore presents a decoding problem, as the space of possible exposure histories which could lead to an observed antibody landscape is large, and observed antibody titres are highly variable due to within-host and laboratory-level effects. Although inferences which account for these mechanisms have provided rich insights into individual-level life course immune profiles, most attempts have been in relatively small cohorts or using small panels of influenza strains,

limiting the conclusions which can be drawn about population-level influenza epidemiology [13,36,37].

Here, we applied an infection history inference method to data from a large serosurvey to reconstruct lifetime individual infection histories and population-level incidence of A/H3N2 influenza in Guangzhou, China [35,36,38]. Infection histories were inferred based on individual-level antibody profiles to a panel of 20 influenza A/H3N2 strains representing viruses that first circulated from 1968 onward. The study population comes from a range of age groups, social backgrounds, and geographical areas, thereby providing an ideal dataset to investigate predictors of influenza infection and small-scale spatial variation. In fitting the model, we also obtained parameter estimates for the underlying antibody kinetics model which allows us to elucidate long- and short-term antibody dynamics. Together, these results provide detailed spatiotemporal insights into the historical epidemiological and immunological dynamics of influenza A/H3N2.

## Results

### Description of participant data

We measured 67,683 HI titres against 20 A/H3N2 strains isolated between 1968 and 2014 at 2 to 3 year intervals from serum samples collected between 2009-12-22 and 2015-06-02 (S1 Fig) as part of the Fluscape cohort study (1,130 individuals, 2 samples each) (S2 and S3 Figs). Sampling was done over 4 study rounds, with a mean time between serum sample collection of 3.87 years (standard deviation: 0.780; 95% quantiles: 1.68 to 4.74 years). This cohort covered 40 unique locations and 651 unique households in a 60 kilometre transect from Guangzhou, China. Participant ages at the most recent sampling round ranged from 6 years to 97 years, with a median of 50 years. Vaccination rates were low in this cohort (S1 Table), consistent with low reported vaccine coverage rates in mainland China, particularly in older individuals [39,40]. We refer to an individual's set of antibody titres against all strains in the panel as their antibody profile. All individual antibody profiles and observed changes in titre are shown in S20 and S21 Figs. Summary statistics of these profiles and full study details have been described elsewhere [38,41].

### Antibody titres vary by age and in space

We saw 5 broad patterns of seropositivity (HI titre $> = 1{:}40$) when stratifying the antibody data by age group and strain (Fig 1). First, individuals had mostly low or undetectable titres against strains that circulated before they were born (cells below the black lines in Fig 1), though many individuals were seropositive against the strain which circulated in the years immediately prior to birth. Second, in the youngest age group (0 to 10 years old), many individuals were also seropositive to pre-birth strains that circulated further back in time (A/California/2004, A/Fujian/2002, A/Fujian/2000, and A/Victoria/1998), indicating the presence of cross-reactive antibodies as these individuals could not have been exposed to those strains (Fig 1C). Third, seroprevalence tended to be high among individuals who were young when a strain was first isolated (cells just above the black line in Fig 1) compared to individuals who were older at the time that the strain was first isolated. Fourth, the total proportion of individuals who seroconverted between sample collection dates was high against recent strains at nearly 50% across all age groups. Finally, some strains exhibited systematically higher titres than others, for example, titres against A/Fujian/2002 and A/Mississippi/1985 were higher than all other strains and particularly high for individuals who were under 10 years old at the time of their first circulation.

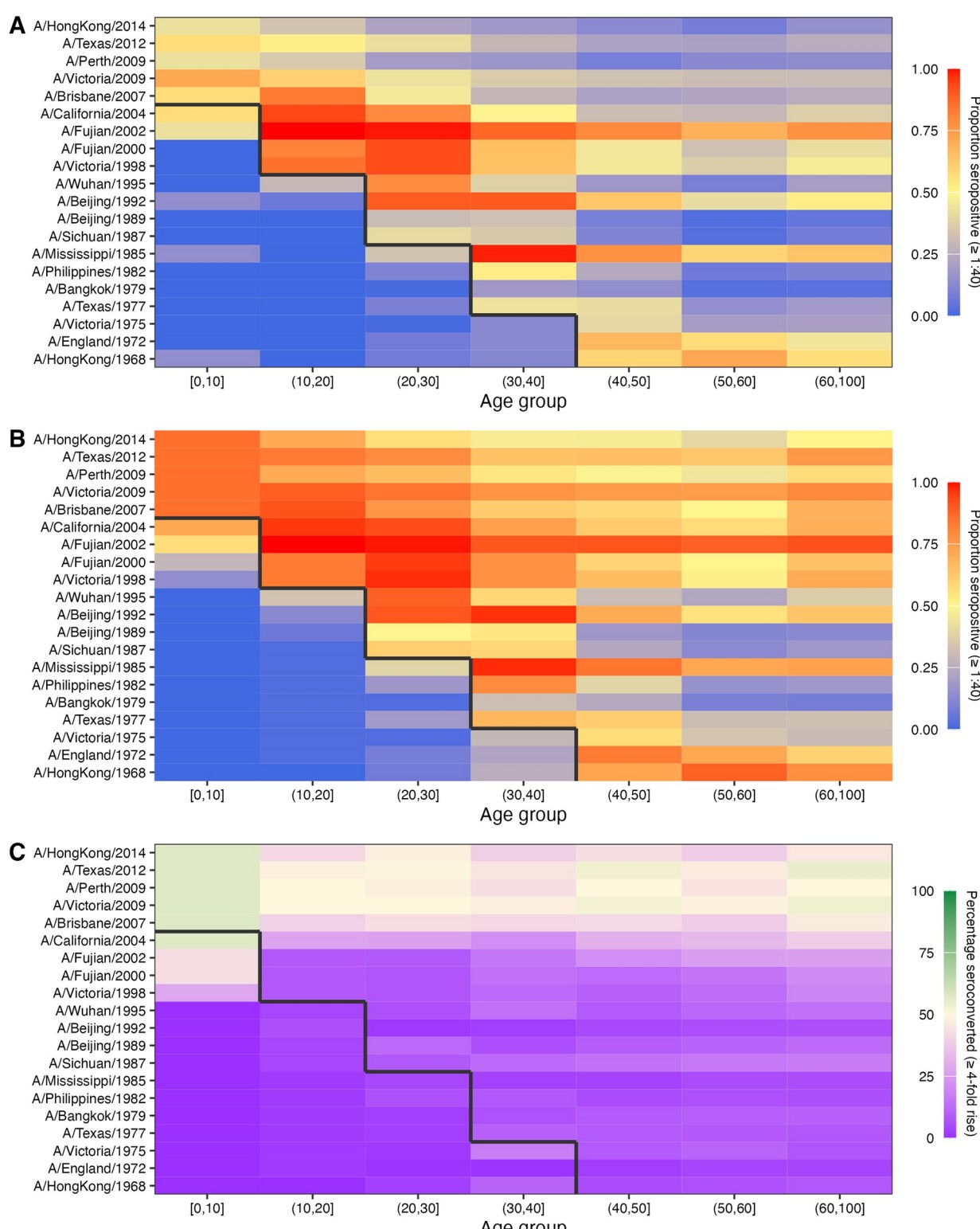

**Fig 1. Proportion of individuals seropositive and seroconverted to 20 A/H3N2 strains circulating from 1968 to 2014 stratified by age.** Solid black line divides age groups that were alive or not at the time of strain circulation. Seropositivity was defined as having an HI titre of ≥1:40 (a log titre of 3). (**A**) First serum sample. (**B**) Second serum sample. (**C**) Seroconversion between samples, defined as a ≥4-fold increase in HI titre. The data underlying this figure can be found at https://doi.org/10.5281/zenodo.12795911.

## Inferring antibody kinetics and individual infection histories from antibody profiles

To infer the set of A/H3N2 influenza strains each individual was infected with, we further developed and used the *serosolver* R-package to fit an infection history and antibody kinetics model to all individual-level antibody profiles [35]. Briefly, the method finds the combination of influenza strains which an individual is most likely to have encountered conditional on their antibody profile, accounting for cross-reactive, transient antibody boosting, and antigenic seniority arising from repeated exposures to antigenically related strains (see Materials and methods). Individuals can be infected with the strain assumed to be circulating in each time period provided they were alive at the time of circulation. Individuals have distinct infection histories, but the parameters governing post-infection antibody kinetics were assumed to be universal, and individual infection probabilities were assumed to arise from a single population-level infection probability parameter per time period, regardless of location. A crucial component of the model is the observation level, which accounts for the fact that some strains elicit systematically higher or lower titres than others in the HI assay (see S1 Text).

We routinely fitted the same model to both the Fluscape cohort data and to previously published antibody profile data from a cohort of 69 individuals in Ha Nam, Viet Nam (see Materials and methods and [37]), and compared the inferred antibody kinetics and epidemic dynamics. Posterior distributions for the model predicted titres and infection histories compared to observed titres are shown for 5 randomly selected individuals from the Fluscape cohort in S4 Fig, and from the Ha Nam, Viet Nam cohort in S5 Fig. Overall, we imputed a total of 10,558 (posterior median; 95% credible interval (CrI): 10,394 to 10,750) distinct infections (see Post-processing of infection history posteriors in Materials and methods) across all individuals and times from the Fluscape data and 547 (posterior median; 95% CrI: 519 to 574) infections from the Ha Nam, Viet Nam data. We also ran varied scenario analyses using simulated data closely resembling the Fluscape data, demonstrating that our inference system was able to accurately recover infection histories, attack rates, and antibody kinetics parameters under a range of assumptions and model misspecifications (S2 Text).

Although there were no virologically confirmed infections reported in the Fluscape study with which to validate our infection history estimates, individuals did self-report influenza vaccination at each study visit, but only within a window of time rather than on a specific date (e.g., in the preceding calendar year; see Materials and methods). We estimated antibody boosting events (infections) with >25% posterior probability for 68.9% of time windows in which individuals self-reported influenza vaccination (62 of 90 windows from 77 individuals; S22 and S23 Figs), compared to 43.1% (mean of null simulations; 95% quantiles: 32.8% to 51.7%) of randomly selected time windows of the same duration from randomly selected individuals, suggesting that our model was more likely to infer antibody boosting events that coincided with self-reported vaccination than in randomly selected time periods. Vaccination windows in which no vaccination/infection was inferred tended to either have antibody boosts identified soon after the reported vaccination, suggesting either delayed boosting or recall error, or very low antibody titres to recent influenza strains, suggesting either inaccurate recall or no vaccine-induced antibody boosting (S23 Fig).

## Antibody kinetics parameters

We obtained estimates for the antibody kinetics model parameters assumed to underlie the generation of observed influenza antibody titres. We estimated parameter values consistent with an initial, broadly reactive antibody response that decays within approximately 1 year to leave an antigenically narrow, persistent antibody response. These estimates are in line with

**Table 1. Estimated antibody kinetics parameters.** The data underlying this table can be found at https://doi.org/10.5281/zenodo.12795911.

| Parameter | Description | Units | Estimates from Fluscape data (posterior median; 95% CrI) | Estimates from Ha Nam, Viet Nam data (posterior median; 95% CrI) |
|---|---|---|---|---|
| $\mu_l$ | Long-term antibody boosting | log HI units | 1.42 (1.39–1.44) | 1.96 (1.89–2.03) |
| $\mu_s$ | Short-term antibody boosting | log HI units | 2.06 (1.99–2.14) | 2.65 (2.43–2.92) |
| $\sigma_l$ | Long-term cross reactivity | Proportion decrease in boost per unit of antigenic distance | 0.0743 (0.0738–0.0749) | 0.115 (0.112–0.118) |
| $\sigma_s$ | Short-term cross reactivity | Proportion decrease in boost per unit of antigenic distance | 0.000122 (2.79e–06–0.000543) | 0.0267 (0.0222–0.0301) |
| $\tau$ | Suppression | Proportion decrease in boost per successive infection | 0.0311 (0.0287–0.0328) | 0.0427 (0.0391–0.047) |
| $\omega\mu_s$ | Waning rate of the short-term response | log HI units lost per year | 2.08 (1.97–2.22) | 2.01 (1.74–2.27) |
| $\varepsilon$ | Standard deviation of the observation error distribution | log HI units | 0.636 (0.631–0.641) | 1.29 (1.27–1.31) |

previous estimates using a similar method applied to a smaller pilot serosurvey from the Fluscape cohort [36], as well as from model fits to the Ha Nam, Viet Nam cohort data (Table 1). However, there were some differences in the estimated magnitude of the antibody response between the Fluscape and Ha Nam data sets. Estimates from the Fluscape data suggested that infection elicited an antigenically narrow, long-term antibody boost of 1.42 log HI units (posterior median; 95% CrI: 1.39 to 1.44) compared to 1.96 log HI units (posterior median; 95% CrI: 1.89 to 2.03) from the Ha Nam data set, and an additional antigenically broad, short-term boost of 2.06 log HI units (posterior median; 95% CrI: 1.99 to 2.14) compared to 2.65 log HI units (posterior median; 95% CrI: 2.43 to 2.92). The estimated waning rate of the short-term response was similar from the 2 data sets, suggesting that the short-term response took 0.995 years (posterior median; 95% CrI: 0.963 to 1.03) to fully subside based on the Fluscape data and 1.33 years (posterior median; 95% CrI: 1.22 to 1.45) based on the Ha Nam data. Finally, the observation error standard deviation was estimated to be larger for the Ha Nam data set at 1.29 (posterior median; 95% CrI: 1.27 to 1.31) versus 0.636 (posterior median; 95% CrI: 0.631 to 0.641) (posterior medians and 95% CrI), though this is likely attributable to the different assumed observation model and measurement of more influenza strains in the latter data set.

## Inferred historical and contemporary attack rates

By combining all individual-level inferred infection histories, we obtained A/H3N2 incidence estimates in the Fluscape cohort for each 3-month window since 1968 (Fig 2A). The estimated median quarterly sample attack rate was 3.54% (median across all posterior samples; 95% CrI: 3.08% to 4.01%). This corresponded to a median annual attack rate of 19.1% (posterior median; 95% CrI: 17.2% to 20.9%), defined as the proportion of alive individuals who experienced at least 1 infection within a calendar year. This was comparable to estimates from the Ha Nam, Viet Nam data set, which gave an estimated median annual attack rate of 18.6% (posterior median; 95% CrI: 14.5% to 22.9%).

Quarterly A/H3N2 attack rate estimates in the Fluscape cohort varied over time, ranging from a minimum of 0.459% (posterior median; 95% CrI: 0.00% to 2.53%) in Q1-1977 (during the re-emergence of A/H1N1) to a maximum of 64.8% in Q1-1968 (posterior median; 95% CrI: 54.2% to 71.3%) (at the beginning of the A/H3N2 pandemic). The attack rate estimate for Q1-1985 was unusually high (51.8% posterior median; 95% CrI: 0.207% to 63.8%), suggesting that there might be residual bias from systematically higher titres for the A/Mississippi/1985 virus not captured by the antibody kinetics and measurement models. Periods of high and low

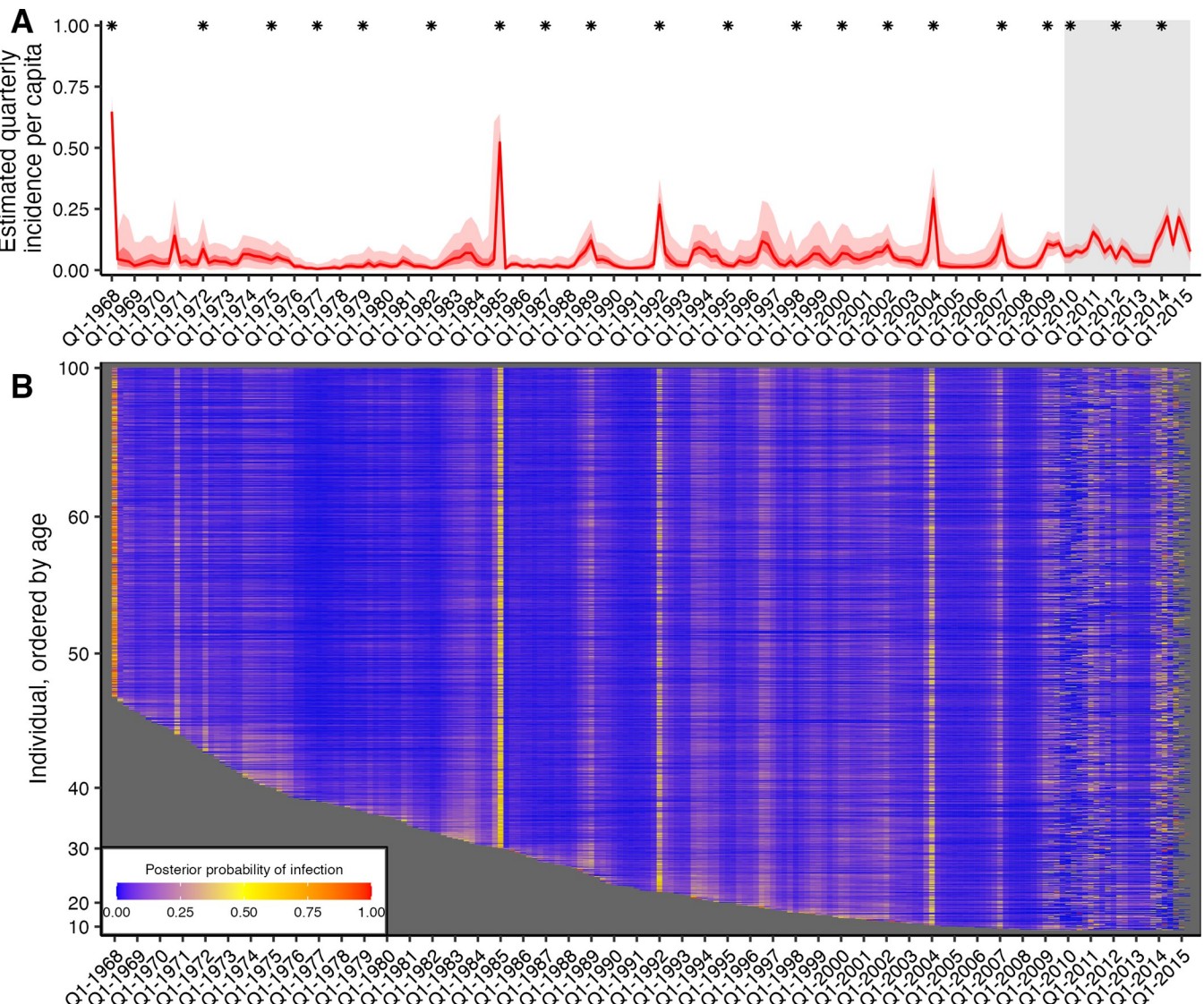

**Fig 2. Quarterly incidence and individual infection histories from the Fluscape data set.** (**A**) Model predicted per-capita incidence per quarter. Attack rates were estimated by dividing the number of inferred infections by the number alive in each 3-month period. Red line shows the posterior median estimate from 1,000 posterior samples. Dark and light red shaded regions show 50% and 95% credible intervals respectively from 1,000 posterior samples. Grey shaded box shows duration of the Fluscape study—the improved precision is due to the inclusion of sera bracketing this time period. Asterisks mark times from which a strain included in the HI panel was first isolated. (**B**) Inferred infection histories for each individual. Each row represents an individual ordered by increasing age in years. Each column represents the time of a potential infection. Cells are shaded based on the proportion of posterior samples with an infection at that time. The grey areas show time periods prior to each individual's birth. The data underlying this figure can be found at https://doi.org/10.5281/zenodo. 12795911.

attack rates were reasonably well synchronised between the Fluscape and Ha Nam data sets, though attack rates were estimated to be significantly higher in Ha Nam from 2000 to 2003, and significantly higher in the Fluscape cohort from 2010 to 2012 (S6 Fig). For example, both data sets gave very high attack rates for 1968, 1989, and 2009, and similar attack rates from 1970 to 1985 (though with substantial uncertainty in the Ha Nam estimates due to the much smaller sample size). There were also periods with clear differences—attack rate estimates were much lower in the Fluscape cohort in the early 2000s, and higher during 2010 to 2012.

**Table 2. Estimated attack rates and infection patterns 2010–2014.** Percentages shown are posterior median and 95% credible intervals. "Attack rate" was defined as the proportion of individuals who were infected at least once in that year. "Seroconverted" gives the percentage of individuals that seroconverted to the measured strain, with ranges showing 95% binomial confidence intervals. "Reinfected" gives the percentage of people that were infected more than once in a year. The data underlying this table can be found at https://doi.org/10.5281/zenodo.12795911.

| Year | Measured strain | Seroconverted (%) N = 1,127 | Estimated attack rate (%) | Reinfected within same year (%) |
|------|-----------------|------------------------------|----------------------------|----------------------------------|
| 2010 | A/Perth/2009 | 47.7% (44.8%–50.6%) | 28.4% (24.8%–32.3%) | 1.68% (0.885%–2.65%)) |
| 2011 | - | - | 41.2% (37.6%–44.4%) | 4.37% (2.85%–5.79%) |
| 2012 | A/Texas/2012 | 50.7% (47.8%–53.6%) | 24.2% (20.3%–28.0%) | 1.53% (0.632%–2.62%) |
| 2013 | - | - | 20.8% (16.4%–25.4%) | 1.21% (0.466%–2.23%) |
| 2014 | A/Hong Kong/2014 | 42.6% (39.7%–45.5%) | 60.8% (57.9%–63.8%) | 5.33% (3.55%–7.27%) |

Quarterly attack rate estimates were substantially higher in the Fluscape cohort after the first serum sample in Q4-2009 than the overall average, with the median quarterly attack rate since Q4-2009 estimated to be 7.79% (median of posterior samples for median per-quarter attack rate; 95% CrI: 3.46% to 21.7%) versus 2.98% before (95% CrI: 0.748% to 13.7%). Annual attack rates (defined as the proportion of individuals who were inferred to have been infected at least once per year) fluctuated, with lower attack rates in 2010, 2012, and 2013, and high attack rates in 2011 and 2014 (Table 2). These attack rate estimates were of a similar magnitude to the proportion of individuals that seroconverted between study visits against strains that circulated during that time. Surveillance data collected from the Guangdong Provincial Centre for Disease Control and Prevention influenza surveillance system during the same time period reported slightly different dynamics, with an increased number of A/H3N2 virus isolates in Q2/Q3-2009, Q3-2010, Q4-2011, and Q1/Q2-2012, as well as a higher proportion of A/H3N2 isolates in 2011 and 2012 [42]. However, direct comparison to our estimated attack rates is difficult, as the influenza surveillance network only collects nasopharyngeal swab samples from patients presenting to sentinel hospitals with ILI, whereas our estimates relate to all exposure events regardless of symptoms.

A substantial proportion of people were estimated to have been reinfected within a single year, with higher reinfection rates in years with higher overall attack rates; 33.9% (posterior median; 95% CrI: 27.0% to 41.0%) of all reinfections between 1968 and 2015 occurred since 2008, whereas only 14.9% of possible infection events were in this time frame, suggesting that reinfections were disproportionately more likely in recent time periods. Annual reinfection rates, defined as having been infected at least once in a year for multiple years, were also high for recent years (Table 3).

## Attack rates varied in space but were not clearly associated with proximity

Previous analyses from this cohort found that antibody titres varied significantly between study locations after accounting for differences in demographics, suggesting that there may be differences in influenza A/H3N2 epidemiological dynamics between locations [1]. We grouped the posterior draws for the Fluscape infection histories by study location to investigate spatial patterns in attack rates. Attack rates exhibited variation between locations and over

**Table 3. Percentage of individuals who were infected at least once per year in 0, 1, 2, 3, 4 or 5 years between 2010 and 2014 inclusive.** The data underlying this Table can be found at https://doi.org/10.5281/zenodo.12795911.

| Number of years with at least one infection between 2010 and 2014 inclusive | 0 | 1 | 2 | 3 | 4 | 5 |
|---|---|---|---|---|---|---|
| Percentage | 12.4% (11.4%-13.5%) | 33.5% (31.4%-35.7%) | 34.0% (31.9%-36.1%) | 15.9% (14.1%-18.0%) | 3.81% (2.83%-4.87%) | 0.265% (0.000%-0.6228%) |

time (S7 Fig), though the timing of high attack rate periods was synchronised. S8 Fig shows snapshots from a video of attack rates over time across the study region, demonstrating that the periods of high influenza incidence are similar across the study locations, but that there is some variation in the timing and magnitude of incidence (full video in S1 Video). The overall coefficient of variation (CoV) for per-quarter attack rates across all locations and times was 1.12 (posterior median; 95% CrI: 1.04 to 1.19). This was reduced to 0.493 (posterior median; 95% CrI: 0.458 to 0.535) after aggregating the infection histories into per-year attack rates, defined as the proportion of individuals infected at least once in a calendar year. There was a strong negative correlation between the estimated posterior median CoV and per-quarter attack rate (Pearson correlation coefficient of −0.829).

To understand if this variation reflects epidemiological differences between locations or simply sampling variation, we generated a comparable null simulation where no significant spatial variation in incidence would be expected (see Spatial correlation in inferred attack rates in Materials and methods). The mean CoV for these simulations was 0.300 (95% quantiles: 0.0307 to 1.22). Given the overlapping uncertainty intervals of the model estimates with the simulations, we cannot exclude the possibility that the estimated attack rates from the Fluscape data exhibited no more variation overall than would be expected by chance if all attack rates were drawn from the same binomial distribution. S9 Fig demonstrates that this pattern is maintained across time, though with higher variation in time periods with low infection rates. Fitting a spatial nonparametric correlation function—a model describing correlation of different spatial units over time as a function of their distance to each other—revealed high correlation in attack rates between locations which showed no clear association with increasing distance from one another (S9D Fig). This consistent correlation across space was also observed when subsetting attack rates by recent (from Q1-2009 onward) or historical times (from Q1-1968 to Q1-2009), when considering either per-quarter or per-year attack rates, and also when considering the proportion of individuals in a population who seroconverted against all or only recent strains.

## Age-specific infection patterns

Periods of high infection probability were largely synchronised across all individuals regardless of age, though individuals typically experienced more frequent infections in the years immediately following birth (S10 Fig). Two age-specific patterns emerged. First, almost all individuals who were alive in 1968 were almost certainly infected in or around 1968, demonstrated by the high posterior probability of infection across all individuals alive at that time. Second, the posterior probability that an individual was infected soon after birth was consistently high, demonstrated by the lower edge of the heatmap in Figs 2B and S10. The posterior mean, median, and 95% CrI on the age of first infection was 1.36, 0.75, and 0.00–6.00 years, respectively (estimates using 1,000 posterior samples for all individuals born since 1968). We note that although these augmented infection histories suggest most individuals were infected soon after birth, our model requires a number of assumptions regarding cross-reactivity to pre-birth strains and antigenic distance between strains which limits our ability to draw this conclusion, particularly given contrasting previous findings from longitudinal serological studies in children [43].

We calculated each individual's age at the time of each infection and found that the number of infections per 10 year period decreased through childhood and became stable in adulthood (Figs 3C and S11). Overall, individuals were estimated to be infected 2.12 times per 10 year period (posterior median; 95% CrI: 1.06 to 7.86), in line with previous estimates (Fig 3D) [36]. Infection frequency patterns and trends with respect to age were similar under different

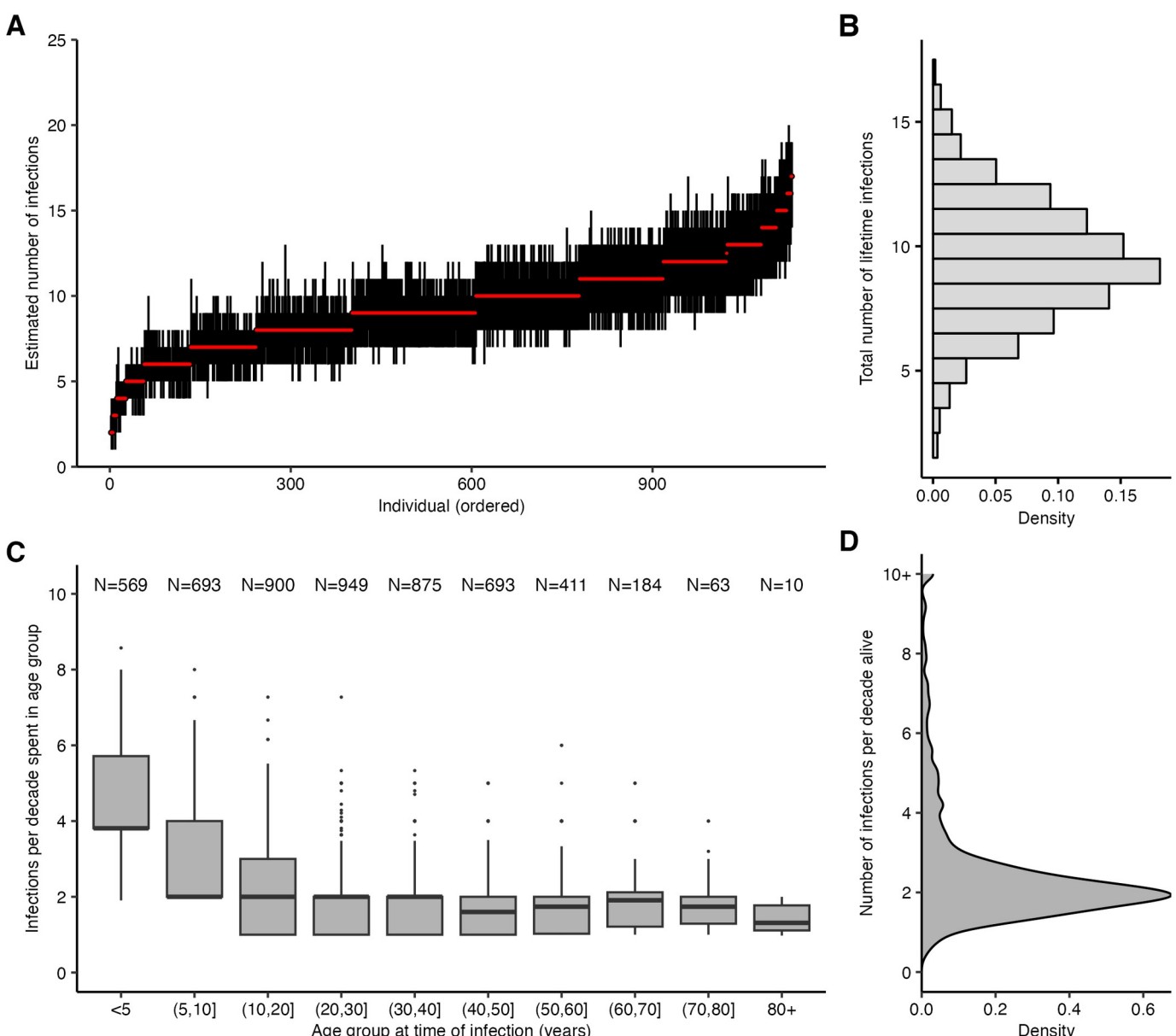

**Fig 3. Age-specific patterns of infection.** (**A**) Pointrange plot shows posterior median and 95% CrI on the total number of lifetime infections for each individual in the Fluscape cohort, ordered by increasing age at time of sampling. (**B**) Distribution of the total number of infections across all individuals based on the posterior median total number of infections. (**C**) Posterior median number of infections per 10 year period stratified by age group at the time of infection, excluding individuals who spent less than 2 years in that age group and including only time periods prior to the first serum sample in Q4-2009 (see S11 Fig for explanation and comparison using all time periods). Text shows sample size within each age group—note this does not sum to the number of individuals in the sample, as individuals contribute to multiple age groups during their lifetime. (**D**) Distribution of individual posterior median number of infections per 10 years alive. The data underlying this figure can be found at https://doi.org/10.5281/zenodo.12795911.

assumptions for the infection history model, suggesting that these findings were driven by features of the data and not an artifact of the model structure (S12 Fig; [35]).

## Relationship between titre and probability of infection

Our model fits estimated not only infection histories, but also each individual's expected HI titre against all strains at each point in time, and thus we were able to estimate the relationship

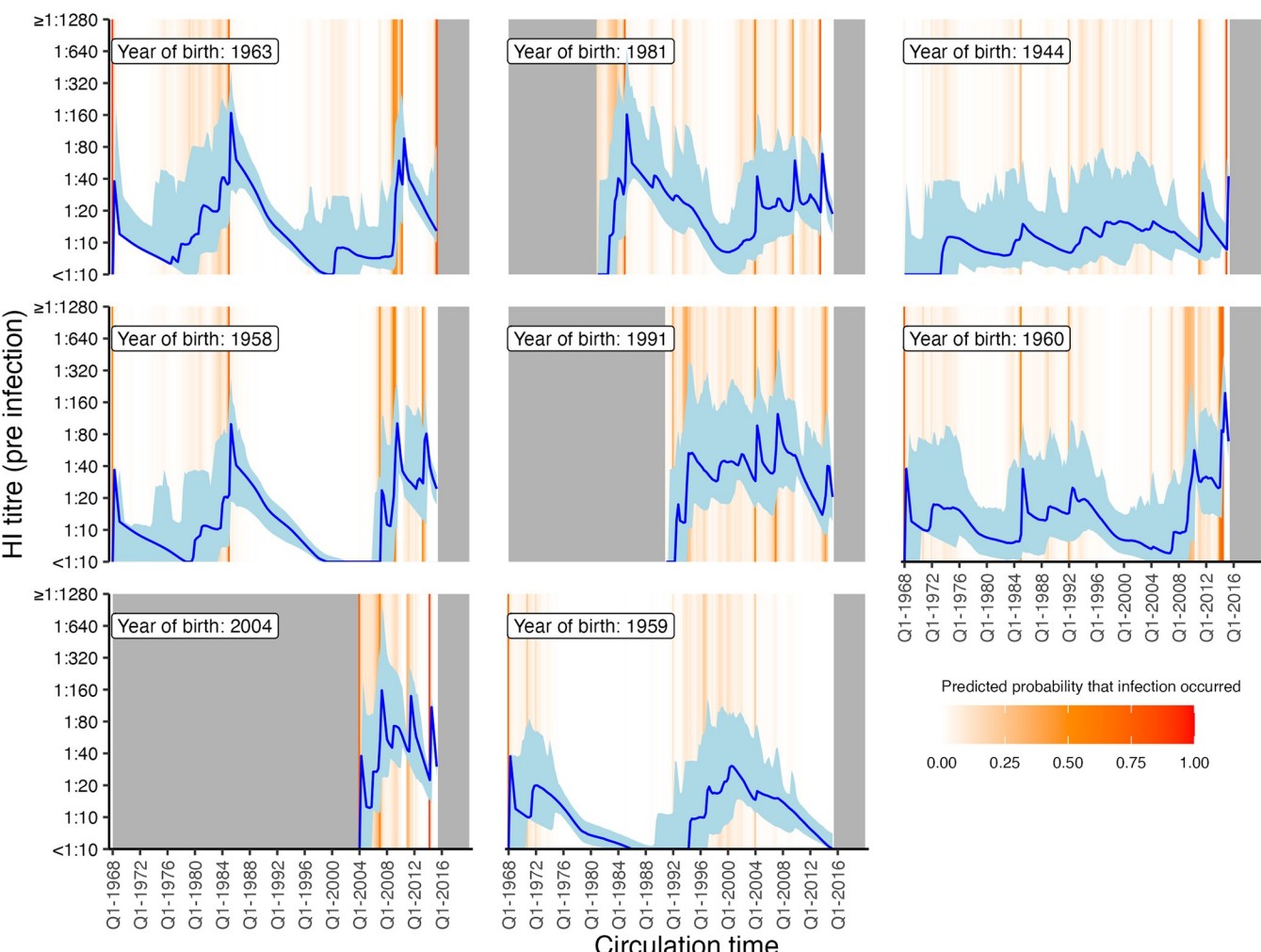

**Fig 4. Model predicted titres against circulating strains since birth.** Each subplot shows one randomly selected individual. X-axis shows time since birth. Blue line and shaded region show model-predicted, true latent titre against the strain assumed to be circulating at each time period (posterior median and 95% CrI). Note that titres are continuous and represent latent, true values, not observations. Orange lines indicate times of high posterior probability of infection. Grey regions show times before birth and after the last serum sample. The data underlying this figure can be found at https://doi.org/10.5281/zenodo.12795911.

between probability of infection and model-predicted latent HI titre (Fig 4). For each sample from the posterior, we found the proportion of time periods (across all times and individuals) where infection was estimated to have occurred, stratified by log HI titre against the circulating strain in that time period. There was a clear pattern of decreasing relative risk of infection as a function of increasing titre (Fig 5). An HI titre of 1:40 (log titre of 3) corresponded to a risk of infection of 0.559 (posterior median; 95% CrI: 0.514 to 0.605) relative to individuals with no detectable titre, consistent with prior evidence from deliberate infection experiments [44]. This pattern appeared to vary with age. An HI titre of 1:40 gave a relative risk of infection of 0.374 (posterior median; 95% CrI: 0.312 to 0.427) in the 0 to 10 age group, but only 0.789 (posterior median; 95% CrI: 0.638 to 0.955) in the 60+ age group.

## Discussion

Influenza A/H3N2 infection histories and attack rates exhibited substantial variation across time and locations in a cohort of 1,130 individuals around Guangzhou, China. We considered

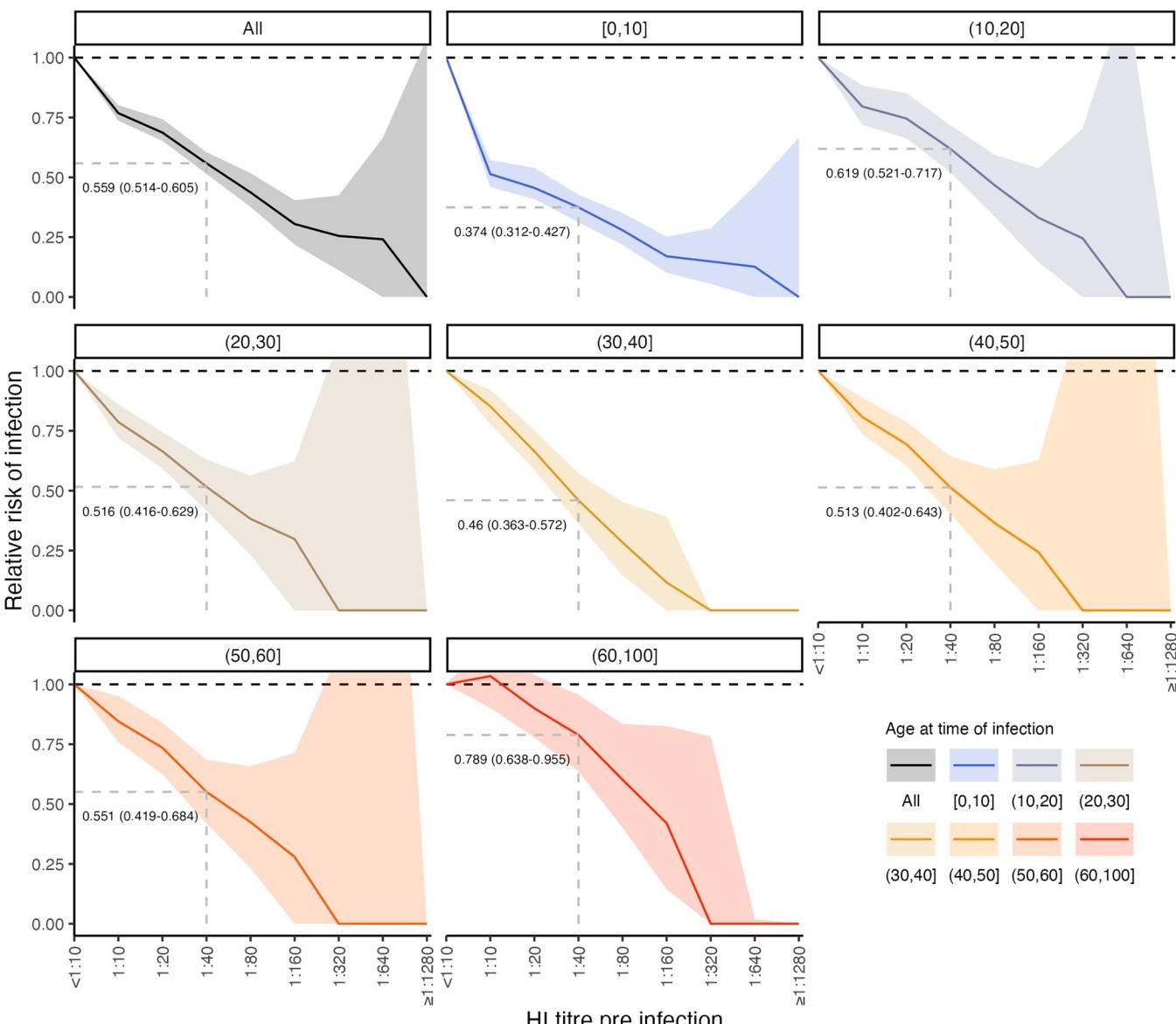

**Fig 5. Estimated relationship between HI titre and probability of infection.** Top left panel shows the relative risk of infection at all time points stratified by model-predicted HI titre against the circulating strain just before infection for all age groups. Remaining plots show the same relationship but stratified by age group in 10-year bands at the time of infection. Solid lines and shaded regions show posterior median and 95% CrI. Note that the uncertainty intervals reflect uncertainty in the imputed infection states and latent antibody titres; the relationships shown here are empirically calculated from the *serosolver* estimates. Wide uncertainty intervals at higher titres reflect limited data as few individuals reach such high titres; the posterior median for number of individuals with HI titre of 1:640 or greater at the time of infection was less than 10 for all but the youngest age group. Vertical dashed line shows HI titre 1:40. Horizontal grey dashed line shows 50% protective titre. The data underlying this figure can be found at https://doi.org/10.5281/zenodo.12795911.

that variation in influenza antibody titres may be generated in 3 ways: (i) exposure to different combinations of viruses at different times; (ii) time-dependent antibody kinetics observed at different times relative to an exposure; (iii) random and strain-specific, systematic variation in the HI assay. We showed that accounting for these mechanisms in a modelling framework allowed for the reconstruction of each individual's complete A/H3N2 infection history since birth conditional on their antibody profile. We reconstructed population-wide and location-specific historical infection, or, more precisely, seroincidence rates from these infection

histories, finding that influenza infection incidence may be higher than suggested by routine surveillance. Also, estimates of each individual's true antibody titre against circulating strains for each 3-month period since birth were generated, showing that elevated antibody titres were associated with a substantial reduction in infection risk that became less effective with increasing age.

We estimated that the incidence of influenza A/H3N2 infections, or at least detectable sero-logical responses, in both the Fluscape cohort and a smaller data set from Ha Nam, Viet Nam was around 19% on average per calendar year since 1968. Results from urban and rural South African communities (the PHIRST cohort) demonstrated similarly high influenza incidence rates under routine RT-PCR testing for influenza A or B [45]. Similarly high incidence rates based on either anti-haemagglutinin or anti-neuraminidase seroconversion were also detected in New Zealand (the SHIVERS cohort) [46]. Our results and those from the PHIRST and SHIVERS studies are at the higher end of annual influenza incidence rate estimates [46–48] and suggest that some individuals may be infected with the same subtype multiple times within a year. Our annual seroincidence rate estimates were particularly high during the Fluscape study period in 2010 to 2014, ranging from 20% to 61% of individuals infected at least once per year, though the cutoff of an influenza season is not as clearly defined for this region as it is for temperate regions [2,49]. These values are higher than previous estimates based on serolog-ical data from Hong Kong for the same time period at 7% to 19% [34,50,51]. This increased incidence during recent time periods could reflect the changing antigenic diversity of influenza A/H3N2 during this time, with the emergence of multiple antigenically distinct clades provid-ing an increased force of infection from strains with varying levels of immune escape leading to more frequent infections [52–54]. Our model is limited in capturing this changing epidemi-ology, as it cannot account for co-circulation of competing strains at the same time, and our HI panel did not have strains representing these multiple clades. However, if individuals in this cohort were infected with strains that were not measured in the HI panel it is likely the model would underestimate infection incidence; antigenic mismatch between the infection strain and the measured strain would lead to lower observed titres, which will lead to a lower posterior probability of infection in the model.

The estimated annual attack rates were similar between the 2 cohorts, but not for all years. A possible driver of these differences is climate, despite these 2 locations being relatively close: Ha Nam, Viet Nam has a tropical climate with no clear influenza seasons [55], whereas Guang-dong, China has a subtropical climate with clear peaks of influenza typically during the sum-mer months, though with peaks in winter/spring in recent years [56–58]. It is possible that differences in climate and behaviour partly explain differences in seasonal dynamics between the 2 locations [59–61], though past studies have been unable to reliably compare overall dis-ease burden between locations due to differences in influenza case detection and reporting without further modelling [62]. Differences in influenza epidemiology between Ha Na, Viet Nam and Guangdong, China could be due to a combination of multiple factors including con-tact rates, climate effects on virus survival and behaviour, immunological history, and domi-nance of different A/H3N2 strains over time.

Reconstruction of lifetime infection histories from this large study population covering all age groups revealed that the frequency of influenza A/H3N2 serological responses is initially high and decreases with age and then remains stable at around 2 infections per decade through most of adult life. The pattern is consistent with previous findings from age-stratified epidemi-ological data [8,34,45,46,48,63], regression analyses of randomized control trial data [64], theo-retical models incorporating age-specific differences in social behaviour [65,66], and data from the Fluscape pilot study [7]. Reassuringly, the clear age patterns we find here were maintained under different assumptions for the infection history model, suggesting that these patterns

emerge from information in the data rather than through the model structure. However, it is important to note that we have estimated the incidence of detectable antibody responses and not necessarily clinically relevant infections—not all infections lead to seroconversions and not all antibody boosts reflect detectable virus shedding [67,68]. Similarly, although our inference method did not distinguish between vaccination and infection, vaccination coverage in this cohort was very low and thus this assumption is unlikely to bias our infection history estimates substantially.

In lieu of virologically confirmed influenza infections with which to validate our model, we used self-reported vaccination to approximate true antibody boosting events, finding that our model was more likely to impute antibody boosts during self-reported vaccination windows than in randomly selected time periods with no reported vaccination. Although this provides some support for the validity of our approach, self-reported vaccination status is not always an accurate measure of true vaccination status due to recall bias, and our model does not account for different antibody kinetics following infection and vaccination. Indeed, for most of the self-reported vaccinations where our model found limited evidence for an antibody boosting event showed limited antibody responses against recent strains; both of these limitations make it difficult to evaluate the validity of our approach based on vaccination data alone.

Although the spatial dynamics of influenza at a large scale are fairly well understood through the use of molecular data, explaining spatial dynamics at smaller scales has revealed contrasting results and remains the focus of ongoing work [12,18,22,69]. Comparable spatio-temporal analyses of seasonal ILI incidence surveillance data in Norway, Sweden, Denmark, and the USA—regions with clearer ILI seasons than Guangdong—showed high spatial correlation in epidemic phase timing and amplitude between locations thousands of kilometres apart that declined with distance [70]. Our estimates show a high correlation of attack rates in space that did not change as a function of distance at a small spatial scale (less than 60 km), suggesting that an individual's life-course of seasonal influenza A/H3N2 infections is largely determined by when they were born and the epidemiology of their wider region, rather than precisely where they live. However, our results are limited as we do not account for the fact that individuals move over their lives, and thus their location recorded at the time their serum sample collected may not match their location earlier in life.

We inferred a negative correlation between log HI titre at the time of exposure and probability of seroresponse in line with the historic deliberate infection data of 50% protection for an HI titre of 1:40 [44,71–73]. We found that this relationship became weaker with age, consistent with recent work showing that although HI titre is a good correlate of protection in children, it is less robust in adults [8]. Non-HI-mediated protection, such as non-haemagglutinin head targeted antibodies or cellular responses, are likely to explain a greater amount of variability in immunity as individuals age [46,74]. The magnitude of our estimates do contrast with other work, which has found that the 50% protective titre threshold may be higher than 1:40 in young children [75,76]. As stated by Hobson and colleagues in 1972, care must be taken in assigning causality to the titre-mediated infection risk estimated here. In the present model, titres necessarily decreased over time following infection due to antigenic drift and short-term waning. If protection is governed by non-HI immunity that wanes at a similar rate, then the same association between titre and relative risk could be observed. Non-HI protection may also explain the findings of decreased infection frequency at older age despite decreasing titre-mediated protection, for example, via non-haemagglutinin head targeted antibodies or cellular responses [46,74]. We note that our estimates are only of relative rather than absolute risk of infection; predicting the future probability of infection for an individual would additionally require knowing the force of infection (probability of exposure) as well as knowing which strain will circulate.

Finally, we generated estimates for post-exposure antibody kinetics from 2 serological data sets. There were a number of similarities and differences in the estimated antibody kinetics parameters based on the Fluscape and Ha Nam cohort data sets. Although differences in tested strains and between-lab protocols limits direct comparison [77], both data sets gave qualitatively similar estimates suggesting a transient, antigenically broad antibody boost that waned within 2 years following exposure to leave a persistent, antigenically narrow antibody boost. The antigenic breadth of the short-term boost was broader in the Fluscape estimates, though this may be due to the inclusion of fewer recent strains and more older individuals than in the Ha Nam cohort, requiring the model to give greater weighting to boosting of historical strains (back-boosting) [37]. We estimated that the transient arm of the seroresponse waned within 1 year based on the Fluscape data compared to 1.33 years based on the Ha Nam cohort data, which is broadly in line with previous observations of antibody titres waning to near baseline approximately 1 year post vaccination [37,78–80]. Overall, this supports previous work suggesting that accounting for sub 4-fold rises in titre, issues arising from non-bracketed sera, and measurement effects may lead to greater sensitivity in identifying infections from paired titres [33,81,82].

Our study has a number of limitations. A key challenge underpinning almost all efforts to analyse quantitative antibody titre data is the variation in titres arising from laboratory processes rather than underlying epidemiology and immunology [77]. Although we adjusted our observation model to account for systematically lower or higher observed titres to particular strains, which may lead to under- or overestimation of attack rates, unaccounted for biases may still remain. External validation or estimation of the offset terms would be useful, but is difficult to do across studies given variability in laboratory protocols and serum potency [77]. We performed paired tests of the serum samples from the 2 study visits to minimise batch effects, though it is possible that the increase in titres and broader reactivity between study rounds (S2 and S3 Figs) might reflect systematic measurement bias rather than increased recent infections as inferred by our model. For example, the delay between sample collection and testing for visit 1 samples might have led to some decline in antibody concentration within the samples. Also, our data were limited in their strain coverage, as we measured titres to only 1 strain for each approximately 2-year period and assumed that that A/H3N2 antigenic evolution followed a smooth rather than punctuated trajectory through antigenic space, and thus we may infer infections with antigenic variants that an individual has never been exposed to [3]. As a sensitivity analysis (not shown), we did attempt to fit the *serosolver* model instead assuming punctuated changes through antigenic space, but we were unable to produce converged model fits due to the discretization of the parameter space and thus we do not present these results. However, we did explore the impact of this potential model misspecification using simulation-recovery experiments, and found that fitting a model assuming continuous antigenic changes over time when data were generated under a punctuated antigenic evolution model did not substantially bias antibody kinetics parameter or attack rate estimates, with the exception of slightly overestimating the short-term antibody waning rate (S2 Text). These assumptions about antigenic evolution, in addition to our simple way of modelling cross-reactive antibody boosting, might explain why our estimates for the age at first infection were much younger than found in serological data from longitudinal cohort studies of children, which are more reliable for estimating the precise timing of these early childhood infections [43]. They may also partially explain why the inferred seroresponse rates did not strictly align with timings of high incidence based on viral isolate data in Guangdong and Hong Kong for the same time period [42,50].

Another limitation is the simplification of the antibody kinetics model used here. The assumption of a fixed-effect term on boosting and waning for all individuals and infection

events masks a substantial amount of individual-level and strain-level variation in kinetics which would be better described by random-effects terms. This is a necessary simplification to ensure identifiability of the post-exposure kinetics parameters while simultaneously inferring hundreds of thousands of latent infection states. Similarly, there are a number of immunological mechanisms, such as titre-dependent boosting and titre ceiling effects which we did not include in the model due to identifiability issues [37,83]. Our model is also limited in explaining back-boosting, as we assumed that cross-reactivity extends linearly across antigenic space. In the short-term antibody boosting arm, cross-reactivity was found to boost the entirety of antigenic space regardless of an individual's age to account for the clear back-boosting of titres against strains encountered early in life. However, this model may be inappropriate for younger individuals, whose immune systems may not have encountered any of these historical antigens and would therefore have no targeted memory B-cells to stimulate. A model that distinguishes boosting of heterologous antibodies through back-boosting of the memory response as opposed to cross-reactive antibodies from a de novo response through targeting shared epitopes would provide a more realistic model of the observed, antigenically broad short-term boost. Finally, we did not model the kinetics of maternal antibodies, which typically wane within the first year of life [84]. The youngest individual in our dataset was 6 years old at the time of sampling, and thus all maternal antibodies would have waned by the time of observation and therefore did not contribute towards the observed antibody titres.

Antibody landscapes based on traditional HI assays, as well as multiplex antigen arrays and deep mutational scanning data, are useful tools for understanding how immunity develops following repeated infection and vaccination to antigenically related viruses [85–90]. The approach and results shown here demonstrate how these antibody profiles can be used to reconstruct lifetime infection histories at a fine spatial scale, providing a new source of augmented data with which to understand long-term epidemiological trends for influenza and other antigenically variable pathogens such as SARS-CoV-2 [91–93].

## Materials and methods

### Ethics statement

Study protocols and instruments were approved by the following institutional review boards: Johns Hopkins Bloomberg School of Public Health, University of Hong Kong, Guangzhou No. 12 Hospital, and Shantou University. Written informed consent was obtained from all participants over 12 years old, and verbal assent was obtained from participants 12 years old or younger. Written permission of a legally authorised representative was obtained for all participants under 18 years old.

### Cohort description—Fluscape

Influenza haemagglutination inhibition (HI) titres were obtained from a previously described cohort in Guangzhou, China, called the Fluscape study (described in detail in a cohort profile [38]) [41]. The Fluscape study is a serological, contact, and demographic survey in and near Guangzhou, China. The study covers 40 locations randomly selected from a 60 km cone-shaped transect extending from Guangzhou city center into the surrounding rural area. Latitudes and longitudes of each study location were assigned based on a central place (e.g., a street or village committee center). A total of 60 households within 1 km of each chosen location were randomly selected and contacted one-by-one until 20 households with at least 1 member willing to provide a blood sample and answer the survey questionnaire were contacted. The majority of locations were classified as rural (30/40), were between 20 and 80 min travel time from Guangzhou city center, and had a population density between 256 and 367,346 persons

per 9 km grid cell. All household members aged 2 or above were eligible to participate, and 5 ml of blood was taken for each blood sample from each visit. Here, we used data from a subset of serum samples, capturing 1,130 individuals who had 2 serum samples taken and analysed from 2 different rounds of sampling between 2009 and 2015 inclusive. The month of sampling, age of participant, vaccination history, and other socioeconomic covariates were available for all individuals.

## Serological data

HI assays were performed for each sample to measure antibody titres against 20 A/H3N2 strains that circulated between 1968 and 2014 inclusive with approximately 2 year spacing between circulation years (A/Hong Kong/1968, A/England/1972, A/Victoria/1975, A/Texas/1977, A/Bangkok/1979, A/Philippines/1982, A/Mississippi/1985, A/Sichuan/1987, A/Beijing/1989, A/Beijing/1992, A/Wuhan/1995, A/Victoria/1998, A/Fujian/2000, A/Fujian/2002, A/California/2004, A/Brisbane/2007, A/Perth/2009, A/Victoria/2009, A/Texas/2012, A/Hong Kong/2014). The 50% tissue culture infectious dose (TCID50) for each virus was determined using Madin-Darby canine kidney (MDCK) cells, calculated using the Reed–Muench method [94]. Blood samples were kept at 4˚C on the day of collection until extracted sera was frozen at −80˚C until testing. Sera were thawed and treated with receptor-destroying enzyme (RDE) derived from Vibrio cholerae to remove nonspecific inhibitors prior to incubation at 56˚C for 30 min, and then absorbed with Turkey red blood cells to remove substances which might lead to nonspecific agglutination. HI assays were conducted in 96-well microtiter plates with 0.5% turkey erythrocytes using 4 haemagglutination units. Sera from each individual's first and second sample were tested side-by-side on the same plate. Repeat titres were generated for all of the strains tested from the second serum sample on a separate plate (23,686 repeat measurements in total), but were not run for the first sample due to insufficient sample volume. Titres were tested in serial 2-fold dilutions from 1:10 to 1:1,280, with the reciprocal of the highest dilution at which haemagglutination was inhibited recorded as the titre, and undetectable titres recorded as <1:10. The full laboratory protocol is described in [95].

For all analyses, titres were transformed to a $log_2$ scale (i.e., 2-fold dilutions), where $y = log_2(D/5)$, giving log titres between 0 and 8 (undetectable titres were treated as a 0 log titre). Seroconversion between study visits was defined as a 4-fold rise in titre, equivalent to a ≥2 unit increase on the log scale. Seropositivity was defined as having a titre of ≥1:40 (log titre ≥3). Further details on laboratory testing have been described previously [41].

## Summary of model

The overall inference task is to obtain estimates for the joint posterior distribution of antibody kinetics parameters ($\boldsymbol{\Theta}$), infection histories ($\boldsymbol{Z}$) for all $n$ individuals, and the attack rate within each of $m$ possible discrete infection periods conditional on the set of observed HI titres ($\boldsymbol{Y}$). Crucially, only $\boldsymbol{Y}$ is observed, so we must infer (or augment) the values of $\boldsymbol{Z}$ as latent features. Throughout the remainder of the methods, we use capital letters to represent random variables and bold letters to represent vectors of random variables. A detailed description of the inference problem and approach is described in [35], but can be summarised as sampling from the posterior distribution:

$$P(\boldsymbol{Z}, \boldsymbol{\theta}|\boldsymbol{Y}) \propto (\prod_{i=1}^{n} \prod_{t=t_{1,i}}^{t_{max,i}} P(\boldsymbol{Y}_{i,t}|Z_{i,1}, Z_{i,2}, \ldots, Z_{i,j\leq t}, \boldsymbol{\theta}))P(\boldsymbol{Z})P(\boldsymbol{\theta})$$

where $\boldsymbol{\theta}$ is the vector of antibody kinetics parameters that describes the link between $\boldsymbol{Z}$ and $\boldsymbol{Y}$.

The set of times $t_{1,i}$ to $t_{max,i}$ gives the time periods when a serum sample was obtained from individual $i$. $P(Y|Z, \theta)$ is defined by the antibody kinetics and observation model, and $P(\mathbf{Z}) = \prod_{i=1}^{n} \prod_{j=1}^{m} P(Z_{i,j})$ is the infection history prior, described below.

An individual's entire infection history is given as a vector of unobserved binary variables, $Z_i = [Z_{i,1}, Z_{i,2}, Z_{i,j}]$. Each infection event, $Z_{i,j}$, is the outcome of a single Bernoulli trial, where $Z_{i,j} = 1$ indicates that individual $i$ was infected with the strain circulating in discrete time period $j$, $Z_{i,j} = 0$ indicates that they were not. The entire infection history matrix $\mathbf{Z}$ for all $n$ individuals across all $m$ time windows was therefore represented by an $n$ by $m$ binary matrix. We estimated infection histories at a 3-monthly resolution, such that each individual had an unobserved infection state for each 3-month period $j$ since birth. Individuals could be infected from the first quarter after they were born. We did not attempt to impute infection states occurring after an individual's last serum sample. The attack rate is then given as $\frac{\sum_i Z_{i,j}}{N_j}$, where $N_j$ gives the number of individuals alive in time period $j$.

Infections lead to the production of antibodies ("seroresponses") that undergo longitudinal and cross-reactive kinetics. The vector of true, latent antibody titres across all time periods is given as $A_i = [A_{i,1}, A_{i,2}, \ldots, A_{i,j}]$, which is generated from an antibody kinetics process with parameters $\theta$ described below. The process generating measured titres from the latent antibody titres is modelled through an observation level. The vector of observations is given as $Y_i = [Y_{i,1}, Y_{i,2}, \ldots, Y_{i,t}]$. Note that the time index for $Y_i$ is different to $A_i$ and $Z_i$, as observations are only made at a subset of $t$ times, whereas latent infection states and antibody titres must be represented at all $j$ times. $Y_{i,t}$ is also itself a vector, as it contains HI titres against all strains measured from that serum sample. A schematic of the full model is shown in S13 Fig and an example simulated antibody landscape over time is shown in S14 Fig.

Note that the estimated attack rates and infection histories incorporate uncertainty in whether elevated titres against a particular strain result from infection with that strain and/or from cross-reactive antibodies from infection with a different, antigenically related strain. For example, a high titre measurement might be explained by a single infection with that strain (leading to a large boost) or by multiple infections with other, antigenically distant strains (leading to multiple small boosts). The antibody kinetics model accounts for both possibilities by including a homologous antibody boosting parameter and a model for cross-reactive antibody boosting as a function of antigenic distance (see below). This structure also allows us to estimate infection states during years from which we do not have a representative influenza strain in the HI panel. For example, in the time period 2010 to 2014 (Table 2), the model samples possible infection histories for that time period where elevated titres against A/Perth/2009, A/Victoria/2009, A/Texas/2012, and A/Hong Kong/2014 could reflect strain-specific antibody boosting from infections in 2010, 2012, and 2014, or cross-reactive antibodies from infection in 2011 and 2013. Hence, we do not present a single estimate for the most plausible infection history, but rather incorporate this uncertainty into the attack rate estimates by using multiple samples from the posterior for each individual's infection history.

## Antibody kinetics, antigenic map, and observation model

We used an existing deterministic model to describe the generation of observed antibody titres following exposure to an influenza strain [35,36]. The model has 3 components: (i) the antibody kinetics model; (ii) the antigenic map; and (iii) the observation model. The antibody kinetics model describes linear short- ($\mu_s$) and long-term ($\mu_l$) antibody boosting on the log scale immediately following infection. The short-term boost wanes over time such that the remaining short-term boost $j - k$ time periods after infection at time $k$ is given by $\mu_s w(k,j) =$

$\mu_s max\{0, 1 - \omega(j\text{-}k)\}$. Long-term boosting is persistent. In addition to boosting antibodies against the infecting strain, infection also elicits the production of cross-reactive antibodies against antigenically related strains. Cross-reactivity is assumed to decrease linearly with antigenic distance by a factor of $d_l(k, j) = max(0, 1 - \sigma\delta_{k,j})$, where $\delta_{k,j}$ represents the antigenic distance between the infecting strain $k$ and the measured strain $j$. The short- and long-term boost for a given strain is therefore given by $\mu_l d_l(k, j) = max\{0, \mu_l(1 - \sigma_l\delta_{k,j})\}$ and $\mu_s d_s(k, j) = max\{0, \mu_s(1 - \sigma_s\delta_{k,j})\}$, respectively. Finally, antigenic seniority by suppression was included, wherein the full amount of boosting decreased linearly by a proportion $\tau$ for every infection following the first. Boosting was scaled by $s(Z_{i,j}, j) = max\{0,(1 - \tau(N_j - 1))\}$ after each infection, where $N_j - 1$ is the number of previous infections before strain $j$. The full model, $f(A|Z,\theta)$ (where $\theta = \{\mu_s, \mu_l, \omega, \sigma_l, \sigma_s, \tau\}$), for expected titre for individual $i$ measured at time $t$ against strain $j$, $A_{i,j,t}$, is given by:

$$A_{i,j,t} = \sum_{k\in Z_i} Z_{i,k}s(Z_{i,k}, k)(\mu_l d_l(k, j) + \mu_l w(k, j)d_s(k, j))$$

A key component of the model is the antigenic map specifying the antigenic distance between A/H3N2 strains and thus their cross-reactivity measured by the HI assay. That is, infection with strain A generates cross-reactive antibodies which also recognise epitopes on strain B, where the degree of cross-reactivity can be modelled as the antigenic distance between the 2 strains. In the *serosolver* model, antigenic distance between strain $k$ and strain $j$, $\delta_{k,j}$, was given by the Euclidean distance between them on the antigenic map. In theory, we might jointly estimate the antigenic map alongside the other model parameters, but at present this is computationally infeasible and thus we assume a fixed antigenic map for model fitting.

We used the antigenic coordinates of the 20 strains measured in the Fluscape study to represent the strains circulating in each time period (S15 Fig). We fit a cubic smoothing spline with low amounts of smoothing (smoothing parameter = 0.3) through the coordinates to provide a comparable model to previous analyses and to smooth over large jumps in the posterior surface, which greatly aids in model convergence by smoothing over multiple posterior modes (see S2 Text) [36]. We also attempted to fit a version of the model assuming punctuated rather than continuous evolution through antigenic space, placing strains into clusters based on previous analyses of A/H3N2 strains in China [96]. Although this punctuated version of the cross-reactivity model may be more realistic [13], the posterior distribution under this model was multimodal and thus we were unable to produce reliably converged model fits.

We assumed that log HI titres observed at time $t$ were normally distributed with mean $A_{i,j,t}$ and variance $\varepsilon$ as in [35,36], with censoring to account for the upper and lower bounds of the assay. The probability of observing an empirical titre at time $t$ within the limits of a particular assay $Y_{i,j,t} \in \{0,..., q_{max}\}$ given expected titre $A_{i,j,t}$ is given by:

$$P(Y_{i,j,t}|Z_i, \theta) = f(Y_{i,j,t}|A_{i,j,t}) = \begin{cases} \int_{Y_{i,j,t}}^{Y_{i,j,t}+1} g(s)ds & \text{if } Y_{i,j,t} \in \{1, q_{max} - 1\} \\ \int_{-\infty}^{1} g(s)ds & \text{if } Y_{i,j,t} = 0 \\ \int_{q_{max}}^{\infty} g(s)ds & \text{if } Y_{i,j,t} = q_{max} \end{cases}$$

where $q_{max} = 8$ and $g(q) = \frac{1}{\sqrt{2\pi\epsilon}} e^{-\frac{(q-y_{i,j})^2}{2\epsilon}}$, the probability density function of the normal distribution and $\varepsilon$ is the standard deviation.

Fitting the model as described so far to the HI titre data led to systematic under- or overestimation of titres to certain strains (S16 and S17 Figs). We therefore introduced fixed, strain-specific measurement offsets similar to [97]. The procedure for estimating these offsets is

described in S1 Text. In short, we added fixed offsets to each predicted titre as:
$Y'_{i,j,t} = Y_{i,j,t} + \chi_j$, where $\chi_j$ is the measurement offset for strain $j$. These additional offset parameters aim to capture the residual observation error not explained by the estimated infection histories, antibody kinetics, and normally distributed observation error.

## Infection history prior

The choice of prior for the infection history matrix $\boldsymbol{Z}$ is discussed in detail in [35]. The crux of the problem is that the choice of prior $P(\boldsymbol{Z})$ determines not only $P(Z_{i,j} = 1)$, but also the prior distribution of total number of lifetime infections, attack rate in a given time period, and time between infections. As we are interested here in reconstructing historical attack rates, we chose to place a Beta prior on the probability of infection in a given time window (prior version 2 in *serosolver*). The infection history matrix $\boldsymbol{Z}$ is then a Beta-Bernoulli distributed variable such that:

$$P(\boldsymbol{Z}) = \prod_{j=1}^{m} \int_0^1 \left(\prod_{i=1}^{n} P(Z_{i,j}|\Phi_j = \phi)\right) P(\Phi_j = \phi) d\phi = \prod_{j=1}^{m} \frac{B(k_j + \alpha, \beta + n_j + k_j)}{B(\alpha, \beta)}$$

where $B$ is the Beta function; $k_j = \sum_i Z_{i,j}$ is the total number of infections across all individuals during time period $j$, and $n_j$ is the number of individuals that could be infected during time period $j$. Values for $\alpha$ and $\beta$ can then be set to give known priors and variance on the total number of infections as $E\left(k_j\right) = n\frac{\alpha}{\alpha+\beta}$ and $\text{Var}\left(k_j\right) = n\frac{\alpha\beta}{(\alpha+\beta)^2}\left[1 + (n-1)\frac{1}{\alpha+\beta+1}\right]$. When $\alpha = \beta$, the attack rate prior has an expectation of *0.5n*, and the variance may be decreased by increasing $\alpha$ and $\beta$. Here, we set $\alpha = \beta = 1$. This choice of prior also implicitly assumes that the total number of lifetime infections for an individual is binomially distributed with success probability $p = \alpha/(\alpha+\beta)$ and $N = m_i$, where $m_i$ is the number of time periods that individual $i$ could be infected.

## Inference using Markov chain Monte Carlo

All models were fitted using the Markov chain Monte Carlo (MCMC) algorithm implemented in the *serosolver* R package. This is a custom, adaptive Metropolis–Hastings algorithm with alternating univariate normal proposals for the model parameters $\boldsymbol{\theta}$ and custom proposals $\boldsymbol{Z}$ for the infection history states. Step sizes for all antibody kinetics parameter proposals were also scaled automatically during the burn in to achieve an acceptance rate of 0.44 for all parameters. Uniform priors were placed on all antibody kinetics parameters $\boldsymbol{\theta}$ shown in S4 Table. Infection history state proposals were randomly chosen between one of 2 options: (i) select 2 potential infection times 12 time periods apart and swap infection states for all individuals that could have been infected at these times; (ii) randomly select 20% of individuals and for each individual, with 50% probability, either sample new infection state values for all possible infection times, or choose 2 potential infection times 12 time units apart and swap their values.

For the main model results, 5 chains were run for 50,000,000 iterations, with the first 20,000,000 discarded as burn-in, to achieve an effective sample size of >200 for all inferred parameters. Convergence was assessed visually and using the Gelman-Rubin diagnostic criteria ($\hat{R}$) with the *coda* R-package [98]. Note that the huge number of iterations is due to the need to impute nearly *4\*N\*T = 4\*1130\*190 = 858,800* latent binary variables in $\boldsymbol{Z}$ (i.e., the infection state for each individual in each 3-month window since 1968, slightly less as individuals could not be infected prior to birth). The ability for the model to recover ground-truth parameters was explored through simulation-recovery, described in S2 Text.

## Post-processing of infection history posteriors

When fitting to the Fluscape data set, the *serosolver* model occasionally imputed continuous runs of repeated infections in adjacent 3-month windows (e.g., imputing infections in Q1-1968, Q2-1968, and Q3-1968), reflecting either genuine repeat infections or the model explaining titres that were higher than a single antibody boost could explain. These infection runs were relatively rare, but were most common for infection in young children and occasionally for recent time periods (S18 and S19 Figs). Runs of repeat infections were not estimated when fitting to simulated data, suggesting that their occurrence in the Fluscape data represents either genuine repeat infections early in life or the model explaining titres that were higher than a single antibody boost could explain. Although these reinfections may be real, particularly among children [99,100], this may be a limitation of the model, which sometimes explains high antibody titres through multiple infections rather than from a single infection eliciting a large boost. Our pipeline therefore included a post-processing step to count these runs as single infection events starting on the date of the first infection in the run. This reduced the total number of distinct infection events from 11,544 infections (posterior median; 95% CrI: 11,358 to 11,736) to 10,558 (posterior median; 95% CrI: 10,394 to 10,750). The attack rate estimates and reinfection rates from the posterior samples prior to removing the runs of consecutive infections are shown in S2 Table, showing that although the post-processing step only slightly reduced overall attack rate estimates, it substantially reduced the reinfection rate estimates, which were as high as 8% in 2013 prior to removing consecutive infections.

We also carried out an additional data augmentation step when calculating annual attack rates for more recent time periods, as we are unable to infer infection states after the latest serum sample for each individual. For example, if an individual's final serum sample is in Q3-2014 and they have been estimated to have no infections in Q1 or Q2, it is not known if the individual should contribute to the numerator and/or denominator when calculating the annual attack rate for 2014. However, because the model assumes that the per-time infection probability is Beta-distributed, it is straightforward to sample new infection states from the model prior to seeing any antibody data using the probability of infection given by (described in [35]):

$$P\left(Z_{i,j} = 1 | \mathbf{Z_{-i,j}}, \alpha, \beta\right) = \frac{k_j + \alpha}{n_j + \alpha + \beta}$$

Where $\mathbf{Z_{-i,j}}$ gives the infection states of all individuals other than $i$ at time $j$, $\alpha$, and $\beta$ are parameters of the Beta prior (assumed to both be 1), $k_j$ is the number of infections in $\mathbf{Z_{-i,j}}$ and $n$ is the number of alive individuals at time point $j$. For every individual with an unknown infection state following their final serum sample, we resample a new infection state using the above formula for each posterior draw.

## Fitting to longitudinal HI titre data from Ha Nam, Viet Nam

To provide a comparison for the model fits to the Fluscape data, we fit the same model to a publicly available data set from Ha Nam, Viet Nam. This data set consisted of 69 participants, each with between 1 and 6 (inclusive) serum samples taken annually from 2007 to 2012 as described previously [37]. In this cohort, HI assays were performed against a panel of up to 57 A/H3N2 strains isolated between 1968 and 2008, with greater sampling of titres against more recent strains. This represents a comparable data set with different dimensions: a much smaller sample size, but far more titres and samples tested per individual. Unlike the Fluscape data, we only had access to the year of sample collection, and thus we could only estimate infection

histories at an annual resolution. Furthermore, we did not have dates of birth available for the individuals, and thus we assumed that all individuals were born prior to 1968, noting that age signals will still be detected based on the individual's antibody profile. For this analysis, we did not include the strain-specific measurement offsets in the observation model, as there were multiple strains tested for each time period. We ran the MCMC algorithm for 5 chains each for 1,500,000 iterations and discarded the first 500,000 iterations as burn-in, which was sufficient to achieve effective sample sizes of >200 and upper 95% confidence intervals of the $\hat{R}$ values of less than 1.1 for all estimated parameters.

## Spatial correlation in inferred attack rates

To test for patterns of influenza incidence in space, nonparametric correlation as a function of distance was tested using the *Sncf* function from the *ncf* R-package [101], where observations were the quarterly attack rate estimates stratified by location ID (40 locations, 190 observations per location). We fit these spline correlograms to 100 samples from the posterior distribution of attack rates, resampling the 40 locations with replacement for each sample to generate posterior medians and 95% CrIs. This analysis was repeated using estimated attack rates since 1968, since 2009, and pre-2009. To provide a comparable null simulation where no significant spatial variation would be expected, we simulated 40 draws from a binomial distribution with $n = 25$ (i.e., 1,000 individuals across 40 locations) and success probability drawn from a uniform distribution between 0 and 1, and repeated this process 10,000 times to calculate the mean and 95% quantiles of the resulting coefficients of variation.

## Relationship between titre and probability of infection

For the analyses investigating the relationship between titre at time of infection and probability of infection, we drew samples from the estimated posterior distribution of antibody kinetics parameters and infection histories, and calculated model-predicted preinfection latent antibody titres against the circulating strain for each individual at each possible infection time. Predicted titres were converted to integers, with titres $\geq 8$ assumed to be 8 to match the observation process. We then calculated the proportion of time periods where infection occurred ($Z_{i,j} = 1$) (the overall probability of infection) stratified by log titre at the time of infection relative to the overall probability of infection with a log titre of 0. As discussed above, some of the inferred infection histories had runs of infections in consecutive time periods (e.g., $Z_i = [0, 0, 0, 1, 1, 1, 0, 0]$). We removed these consecutive infections from the probability of infection statistics (e.g., *[0, 0, 0, 1, 1, 1, 0, 0]* becomes *[0, 0, 0, 1, 0, 0, 0, 0]*). We repeated this process for 1,000 posterior samples to generate median and 95% CrI estimates on the relationship between titre and relative risk of infection.

## Validation using self-reported vaccination

We validated our inference approach by comparing the model-predicted probability of infection to self-reported influenza vaccination. Vaccination coverage was low in this cohort (S1 Table), and self-reported vaccination status is not a perfect reflection of true vaccination status. Furthermore, vaccination status was reported only within a window of time rather than on a specific date: individuals were asked if they had ever received an influenza vaccination and could give one of 6 answers: never vaccinated; in the same calendar year; in the preceding calendar year; 2 to 5 years ago; 5+ years ago; or unsure. Individuals were also asked if they had been vaccinated since their last study visit. Where an individual reported vaccination in one of these time periods, we define the covered time period as the "vaccination window."

To assess the accuracy of our inference approach, we defined sensitivity as the proportion of vaccination windows +/− 3 months which contained at least 1 inferred infection with more than 25% posterior probability (i.e., we expect vaccination windows to contain inferred infections if our inference is accurate). However, a complication is that these windows are wide—an individual reporting vaccination 2 to 5 years ago has a 3-year wide vaccination window, and thus the individual is likely to have been infected during that time regardless of vaccination. Therefore to provide a comparable null simulation, we took each of the windows of time with reported vaccination and reassigned them to randomly chosen individuals at random times (e.g., if individual 1 reported vaccination between Q1-2013 and Q1-2015; the null simulation reassignment might move the window to individual 2 in Q2-1980 to Q2-1982, regardless of individual 2's vaccination status). We repeated this process 100 times to generate 95% uncertainty intervals for the null simulation. We then compared the proportion of vaccination windows which contained inferred infections based on reported vaccination to the null simulations. If the upper 95% uncertainty interval of the null simulation was lower than the proportion of vaccination windows with inferred infections using the real data, we took this as evidence that our model was more likely to infer antibody boosting events that coincided with self-reported vaccination than in randomly selected time periods.

## Supporting information

**S1 Fig. Distribution of serum sampling times from the Fluscape cohort.** First and second visits refer to an individual's serum sample order, which may differ from the sample collection round of the overall study. The data underlying this figure can be found at https://doi.org/10.5281/zenodo.12795911.
(TIF)

**S2 Fig. Distribution of log HI titres by study location at first serum sample.** Each cell represents the log HI titre for 1 individual measured against 1 strain, shown on the x-axis. Locations were grouped into quintiles based on increasing distance from Guangzhou city center (bottom panels). Individuals were grouped by age and plotted with increasing age. Colours to the left of each subplot show age group. The data underlying this figure can be found at https://doi.org/10.5281/zenodo.12795911.
(TIF)

**S3 Fig. Distribution of log HI titres by study location at second serum sample.** Each cell represents the log HI titre for 1 individual measured against 1 strain, shown on the x-axis. Locations were grouped into quintiles based on increasing distance from Guangzhou city center (bottom panels). Individuals were grouped by age and plotted with increasing age. Colours to the left of each subplot show age group. The data underlying this figure can be found at https://doi.org/10.5281/zenodo.12795911.
(TIF)

**S4 Fig. Example inferred latent antibody titres and infection histories.** (**A**) Model-predicted titres compared to observed HI titres at each sampling time for 5 randomly selected individuals. Rows represent individuals. Subplots show antibody titres based on serum samples taken at that time. X-axis represents a position along the antigenic summary path. Black diamonds show observed titres. Black line and green shaded regions show posterior median and 95% credible intervals (CrIs) on model-predicted latent titres (dark green) and 95% prediction intervals (light green). Orange bars show posterior probability of infection in that 3-month window. Grey rectangles denote the limit of detection of the HI assay. Purple rectangles show time periods before birth. (**B**) Posterior median and 95% CrI for the cumulative number of

infections over time from birth (purple dashed line). The data underlying this figure can be found at https://doi.org/10.5281/zenodo.12795911.
(TIF)

**S5 Fig. Model fit to data from Ha Nam, Viet Nam.** (**A**) Model-predicted titres compared to observed HI titres at each sampling time for 5 randomly selected individuals, as in S4 Fig. Diamonds show titre measurements; green shaded region shows 95% CrI and 95% prediction intervals; dashed line shows posterior median; orange bars show posterior probability of infection in a given time window. (**B**) Posterior median and 95% credible intervals (CrI) for the cumulative number of infections over time from birth (orange). Note that date of birth information was not available for these individuals. The data underlying this figure can be found at https://doi.org/10.5281/zenodo.12795911.
(TIF)

**S6 Fig. Comparison of annual attack rates using data from Ha Nam, Viet Nam and the Fluscape study in Guangzhou, China.** Annual attack rates were defined here as the proportion of individuals who experienced at least 1 infection per year. Yellow shaded regions show time periods where >95% or <5% of posterior samples suggested a greater attack rate in the Fluscape cohort than Ha Nam, whereas green shaded regions show time periods where between 25% and 75% of posterior draws suggested a greater attack rate in the Fluscape cohort. Some time periods showed high uncertainty for the Ha Nam data set, as few individuals in the sample were alive during that time (e.g., 1969–1980). Attack rate estimates were estimated to be higher in the Fluscape cohort from 2000 to 2003 inclusive with more than 95% posterior probability. This might reflect a genuine different in A/H3N2 epidemiology during that time, but may also be partially driven by systematic biases in titre measurements to strains isolated during that time period—the fits to the Fluscape data include a positive offset term for titres against A/Fujian/2002, which leads to lower attack rate estimates in that time period, whereas fits to the Ha Nam data do not. The time period from 2010 to 2012 also shows higher attack rates in the Fluscape cohort but with similar relative patterns. The data underlying this figure can be found at https://doi.org/10.5281/zenodo.12795911.
(TIF)

**S7 Fig. Quarterly attack rates across the 40 Fluscape study locations.** Each row represents 1 study location ordered by increasing distance from Guangzhou city center. Each column represents a 3-month period. Cells are shaded by (**A**) the posterior median inferred attack rate or (**B**) the coefficient of variation of the posterior quarterly attack rate estimate for each location. The data underlying this figure can be found at https://doi.org/10.5281/zenodo.12795911.
(TIF)

**S8 Fig. Distribution of quarterly attack rates by location over time.** Each panel is one frame from a full animation available in S1 Video. Each coloured point shows the inferred attack rate in each of the 40 locations, with size and shading reflecting the posterior median attack rate. Underlying the plot is a map of the study area, with each grid cell shaded by its $log_{10}$ population density. The data underlying this figure can be found at https://doi.org/10.5281/zenodo.12795911.
(TIF)

**S9 Fig. Spatial variation in annual attack rate estimates over time and correlation between nearby locations.** (**A**) Coefficient of variation, (**B**) overall mean, and (**C**) standard deviation of posterior median annual attack rate estimates from the 40 study locations. Solid lines and shaded regions show posterior medians and 95% credible intervals. In (**A**), the solid horizontal

line shows the overall mean coefficient of variation across all time. Dashed horizontal lines show 95% quantiles of simulated coefficients of variation under the assumption that attack rates are the same across space. (**D**) Fitted spline correlograms showing spatiotemporal correlation in attack rates and proportion seroconverted with increasing distance. The first 3 plots show the spline correlogram calculated using the *Scnf* function from the *ncf* R-package for each of 100 posterior samples for the 40 location-specific attack rates. Solid lines and shaded regions show median and 95% quantiles of the predicted covariance function for these 100 samples, coloured by the level of aggregation used to calculate the attack rates. Each subplot shows the same calculation using either attack rates from all times, prior to Q1-2009 or Q1-2009 onwards. For the final plot ("Seroconversion"*),* we calculated the spatial correlation in the proportion seroconverted in each of the 40 study locations, treating strain isolation time as the time variable. Solid lines and shaded regions show median and 95% quantiles of 1,000 bootstrapped observations. We repeated the analysis using either seroconversion to all strains, or only A/Victoria/2009, A/Perth/2009, A/Texas/2012, and A/HongKong/2014. The data underlying this figure can be found at https://doi.org/10.5281/zenodo.12795911.
(TIF)

**S10 Fig. Quarterly attack rates by birth cohort.** Model predicted per-capita incidence per quarter stratified into 5 birth cohorts. Attack rates were estimated by dividing the number of inferred infections by the number alive in each birth cohort in each 3-month period. Solid lines show the posterior median estimate from 1,000 posterior samples. Shaded regions show 95% credible intervals from 1,000 posterior samples. Grey shaded box shows duration of the Fluscape study. The data underlying this figure can be found at https://doi.org/10.5281/zenodo.12795911.
(TIF)

**S11 Fig. Posterior median number of infections per 10-year period stratified by age group at the time of infection, including either all time periods (blue), time periods during the Fluscape study period (Q4-2009 onward; orange) or only time periods prior to the Fluscape study (prior to Q4-2009; grey).** Excludes infection states for individuals who spent less than 2 years in that age group. We present infection rate estimates using only infections from time periods prior to the first serum sample in Q4-2009 in Fig 3C. This is because there are many individuals representing the oldest age group at time of infection for time periods post Q4-2009, but relatively few from pre Q4-2009 (as individuals who were very old in historical time periods are no longer alive). In contrast, younger age groups are better represented across historical time periods (as those individuals are still alive at the time of sampling). Combining this biased representation of older individuals with much higher estimated incidence rates in recent time periods weighs the infection rate estimates for the older age groups much higher simply because most of their infections come from this time period. Therefore, we present age-stratified infection rate estimates using only pre Q4-2009 infections in the main text. The data underlying this figure can be found at https://doi.org/10.5281/zenodo.12795911.
(TIF)

**S12 Fig. Age-specific patterns of infection under an alternative infection history model using estimates from the model version described in S1 Text.** Results shown are identical to those in Fig 3, but assuming that (i) individuals can only be infected once per year (i.e., annual resolution infection histories rather than quarterly); (ii) the infection history model is placed upon an individual's total number of lifetime infections and not their per-time probability of infection (see [35] for further detail on implications of different prior assumptions); (iii) we did not remove runs of continuous infections from the posteriors. (**A**) Pointrange plot shows

median and 95% CrI on the total number of lifetime infections for each individual, ordered by increasing age. (**B**) Distribution of the total number of infections across all individuals based on the posterior median total number of infections. (**C**) Posterior median number of infections per 10-year period stratified by age group at the time of infection, excluding individuals who spent less than 2 years in that age group, and including only time periods prior to Q4-2009. Text shows sample size within each age group—note this does not sum to the number of individuals in the sample, as individuals contribute to multiple age groups during their lifetime. (**D**) Posterior median number of infections per 10 years alive across all individuals. Under this prior, we estimated that individuals are infected 2.08 times per 10-year period (posterior median; 95% CrI: 1.04–7.62). The data underlying this figure can be found at https://doi.org/10.5281/zenodo.12795911.
(TIF)

**S13 Fig. Schematic of the full _serosolver_ model representing a single individual infected with 2 strains, B and H, over a 10-year time period.** (**A**) Example, randomly generated population-level infection probabilities. At the population level, the model describes a per-time-period probability of infection applied to the whole population. These probabilities are used to simulate a vector of latent binary infection states, $Z_i$, for each individual as a series of independent Bernoulli trials (shown as a vector of 1s and 0s). (**B**) The antigenic relatedness of A/H3N2 strains is given by an antigenic map, where the degree of cross reactivity between any 2 strains is given by their Euclidean distance on the map. (**C**) Antibody levels against the infecting strain (strain B) are boosted and wane, given by the summation of transient short-term boosting and persistent long-term boosting. (**D**) Infection with strain B also induces cross-reactive antibodies against all other strains, here showing antibody levels to strain H. The degree of cross-reactivity is proportional to the antigenic distance between the infecting and measured strain. Later on, the individual is infected again, this time with strain H, inducing further antibody boosting and waning. An antigenic seniority parameter, τ, reduces each successive boost as a function of the number of previous infections, *N*. Snapshots of these underlying antibody kinetics are observed through serum samples (blood vials; (**E**) and (**F**)) distributed according to a truncated, discretized normal distribution with standard deviation parameter ε.
(TIF)

**S14 Fig. Simulated antibody landscapes and infection histories over time for one individual using the antibody kinetics model.** Each subplot shows the antibody landscape for that time period. The blue region gives the antibody landscape in that time period, whereas the red region gives the antibody landscape in the preceding time period. The x-axis of each subplot gives the identity of the strain assumed to be circulating in that year. The grey region shows the time period and thus strains that circulated before the individual was born. Vertical dashed lines give the timing/strains the individual was infected with.
(TIF)

**S15 Fig. Antigenic coordinates of measured strains and strains assumed to be circulating in each time period.** Each coloured point shows the location of the labelled strain on the antigenic map given in [37]. Strains which are further apart are less antigenically similar, and therefore exhibit less cross-reactivity following a seroresponse. The smoothing spline shows the inferred coordinates of each strain *j* assumed to have circulated in each 3-month time period, where each black point shows the assumed location in successive time periods. First, a cubic smoothing spline was fitted to the locations of the measured strain with smoothing parameter 0.3. Second, a linear model was fitted to predict the x-coordinate as a function of the strain isolation time. Finally, we generated predicted x-coordinates for each possible *j*

given the circulation time from the linear model, and then used the predicted x-coordinate to predict the y-coordinate from the fitted smoothing spline. The antigenic distance between each pair of strains $k$ and $j$ was then calculated based on their Euclidean distance. The data underlying this figure can be found at https://doi.org/10.5281/zenodo.12795911.
(TIF)

**S16 Fig. Quarterly incidence and individual infection histories from the Fluscape data set without strain-specific measurement offsets.** Identical to Fig 2, but without the inclusion of strain-specific measurement offsets in the observation model. (**A**) Model predicted per-capita incidence per quarter. Attack rates were estimated by dividing the number of inferred infections by the number alive in each 3-month period. Red line shows the posterior median estimate from 1,000 posterior samples. Dark and light red shaded regions show 50% and 95% credible intervals respectively from 1,000 posterior samples. Grey shaded box shows duration of the Fluscape study. Asterisks mark times from which a sample circulating strain was tested. (**B**) Inferred infection histories for each individual. Each row represents an individual ordered by increasing age in years. Each column represents the time of a potential infection. Cells are shaded based on the number of the posterior samples with an infection at that time divided by the total number of posterior samples for that infection state. The data underlying this figure can be found at https://doi.org/10.5281/zenodo.12795911.
(TIF)

**S17 Fig. Distribution of antibody titre prediction errors (observed—model predicted) when fitting the *serosolver* model ignoring strain-specific measurement offsets.** (**A**) Distribution of titre prediction errors stratified by tested A/H3N2 strain. (**B**) Overall distribution of titre prediction errors across all measured viruses. Vertical red line shows $x = 0$; buckets to the right of the red line suggest underestimation of titres; histograms to the left of the red line buckets suggest overestimation of titres. The data underlying this figure can be found at https://doi.org/10.5281/zenodo.12795911.
(TIF)

**S18 Fig. Runs of repeated infections are disproportionately more likely to occur early in life.** (**A**) Proportion of infection episodes which are estimated as runs of consecutive infections by age at start of infection run, using either all infection episodes or only those from individuals born after 1985. (**B**) Number of inferred infection episodes which are estimated as runs of consecutive infections stratified by the infection order in each individual's infection history. For example, if the run is the first infection an individual has experienced, this is given an infection number of one. Of the 10,558 distinct infection episodes, 757 (posterior median; 95% CrI: 676–861) were runs of 2 consecutive infections, 79 (posterior median; 95% CrI: 62–97) were runs of 3 consecutive infections and 19 (posterior median; 95% CrI: 13–27) were runs of 4 or more. The data underlying this figure can be found at https://doi.org/10.5281/zenodo.12795911.
(TIF)

**S19 Fig. Model fits demonstrating typical profiles where consecutive infection runs were imputed.** Model-predicted titres compared to observed HI titres at each sampling time for 2 individuals, as in S4 Fig. Diamonds show titre measurements; green shaded region shows 95% CrI and 95% prediction intervals; dashed line shows posterior median; orange bars show posterior probability of infection in a given time window. Purple rectangles show time periods prior to birth. (**A**) Example of an individual estimated to have experienced multiple consecutive infections immediately following birth to explain high titres. (**B**) Example of an individual estimated to have experienced multiple consecutive infections between the 2 serum sampling

times to explain the drastic increase in titres to recently circulating strains. The data underlying this figure can be found at https://doi.org/10.5281/zenodo.12795911.
(TIF)

**S20 Fig. Observed antibody profiles for all individuals in the Fluscape cohort.** Each subplot shows antibody levels measured against each of the 20 H3N2 strains. The x-axis shows the isolation year of the measured strain. The areas are shaded by sample number, showing titre measurements from the first (blue) and second (red) samples. Grey rectangles mark strains which circulated before that individual was born. The vertical coloured lines show the timing of the serum samples relative to the strain isolation times. Plots where the red region extends above the blue region reflect antibody boosting between the first and second serum sample. Where multiple titres were measured against the same strain from the same serum sample, we plotted the mean of the log titres. The data underlying this figure can be found at https://doi.org/10.5281/zenodo.12795911.
(PDF)

**S21 Fig. Observed changes in antibody titre for all individuals in the Fluscape cohort.** As in S20 Fig, but showing fold-change in titre against each strain between samples. Each subplot shows the change in antibody levels measured against each of the 20 H3N2 strains. The x-axis gives the isolation year of the measured strain. The vertical grey line shows the timing of birth or 1968, whichever was later. Bars are shaded orange to denote antibody boosting and green to denote antibody waning. Horizontal dash lines indicate 2-fold boosting or waning. The data underlying this figure can be found at https://doi.org/10.5281/zenodo.12795911.
(PDF)

**S22 Fig. Comparison of observed antibody profiles, model-predicted infection states, and self-reported vaccination status for accurately detected vaccinations.** Left-hand column shows raw data as in S20 Fig. Right-hand column shows the model-estimated posterior probability of infection (higher grey area suggested higher probability of infection) compared to self-reported vaccination states. Orange regions show time periods in which individuals reported having been vaccinated for influenza. Purple regions show time periods in which individuals reported no vaccination for influenza. Vertical dashed lines show the time of serum sample collection. Individuals were included in this plot (rather than S23 Fig) if the posterior probability of infection during one of the orange time windows was >25%. The data underlying this figure can be found at https://doi.org/10.5281/zenodo.12795911.
(PDF)

**S23 Fig. Comparison of observed antibody profiles, model-predicted infection states, and self-reported vaccination status for missed vaccinations.** Left-hand column shows raw data as in S20 Fig. Right-hand column shows the model-estimated posterior probability of infection (higher grey area suggested higher probability of infection) compared to self-reported vaccination states. Orange regions show time periods in which individuals reported having been vaccinated for influenza. Purple regions show time periods in which individuals reported no vaccination for influenza. Vertical dashed lines show the time of serum sample collection. Individuals were included in this plot (rather than S22 Fig) if the posterior probability of infection during one of the orange time windows was <25%. The data underlying this figure can be found at https://doi.org/10.5281/zenodo.12795911.
(PDF)

**S24 Fig. Comparison of estimated antibody landscapes from the same model fit with and without strain-specific measurement offsets.** Rows represent individuals. Subplots show

antibody titres based on serum samples taken at that time. X-axis represents a position along the antigenic summary path. Black diamonds show observed titres. Black line and blue or green shaded regions show posterior median and 95% credible intervals (CrI) on model-predicted latent titres (dark blue/green) and 95% prediction intervals (light blue/green). Orange bars show posterior probability of infection in that 3-month window. Grey rectangles denote the limit of detection of the HI assay. Purple rectangles show time periods prior to birth. (**A**) Model-predicted titres compared to observed HI titres at each sampling time for 3 randomly selected individuals, as in S5 Fig, but from fitting the model without strain-specific measurement offsets. (**B**) as in (**A**), but with the estimated strain-specific measurement offsets included. The data underlying this figure can be found at https://doi.org/10.5281/zenodo.12795911.
(TIF)

**S25 Fig. Estimated posterior distributions for antibody kinetics parameters (green shaded region) using the fitted model described in S1 Text compared to true values used in the simulation (blue line).** The data underlying this figure can be found at https://doi.org/10.5281/zenodo.12795911.
(TIF)

**S26 Fig. Estimated posterior distributions for strain-specific measurement offsets (green shaded region) using the fitted model described in S1 Text compared to true values used in the simulation (blue line).** The data underlying this figure can be found at https://doi.org/10.5281/zenodo.12795911.
(TIF)

**S27 Fig. Assessment of model fitting accuracy based on simulated data.** Results shown are from fitting the full model to simulated infection histories and antibody titres with known parameters. (**A**) Model-predicted titres compared to observed HI titres at each sampling time for three individuals. (**B**) Posterior median and 95% credible intervals (CrI) for the cumulative number of infections over time from birth (orange). Blue solid line shows the true, known cumulative number of infections. Purple dashed line shows the time of birth. (**C**) Posterior estimated per-capita per-3-month attack rates. Red line and shaded region shows posterior median and 95% CrI. Grey line shows the true values used for the simulation. (**D**) Shaded regions show posterior distributions of estimated antibody kinetics parameters. Dashed lines show the true value used for simulation. Note the x-axis range is small relative to the prior ranges in S5 Table. The data underlying this figure can be found at https://doi.org/10.5281/zenodo.12795911.
(TIF)

**S28 Fig. Assessment of model fitting accuracy based on simulated data when strain-specific measurement offsets are ignored.** Results shown are from fitting the full model to simulated infection histories and antibody titres with known parameters where strain-specific measurement offsets are used in the simulation, but ignored in the fitted model. (**A**) Model-predicted titres compared to observed HI titres at each sampling time for 3 individuals. (**B**) Posterior median and 95% credible intervals (CrI) for the cumulative number of infections over time from birth (orange). Blue solid line shows the true, known cumulative number of infections. Purple dashed line shows the time of birth. (**C**) Posterior estimated per-capita per-3-month attack rates. Red line and shaded region shows posterior median and 95% CrI. Grey line shows the true values used for the simulation. (**D**) Shaded regions show posterior distributions of estimated antibody kinetics parameters. Dashed lines show the true value used for simulation. Note the x-axis range is small relative to the prior ranges in S5 Table. The data

underlying this figure can be found at https://doi.org/10.5281/zenodo.12795911.
(TIF)

**S29 Fig. Antigenic maps used for the simulation (A) compared to the map produced using data from Bedford and colleagues (B), and the map which assumes a punctuated path through antigenic space (C).** Axes show arbitrary antigenic dimensions. Coloured points show the position of individual strains—all labels of the same colour correspond to the same antigenic cluster. Black line shows the fitted antigenic summary path for each map. Black dots show the antigenic position of the strain corresponding to each time period; (**A**) and (**B**) assume continuous evolution through antigenic space, whereas (**C**) assumes that the same position is used for all strains within a cluster. The data underlying this figure can be found at https://doi.org/10.5281/zenodo.12795911.
(TIF)

**S30 Fig. Assessment of model fitting accuracy when the antigenic map used for model fitting does not match the one used for simulation.** Results shown are from fitting the full model to simulated infection histories and antibody titres with known parameters. (**A**) Model-predicted titres compared to observed HI titres at each sampling time for 3 individuals. (**B**) Posterior median and 95% credible intervals (CrI) for the cumulative number of infections over time from birth (orange). Blue solid line shows the true, known cumulative number of infections. Purple dashed line shows the time of birth. (**C**) Posterior estimated per-capita per-3-month attack rates. Red line and shaded region shows posterior median and 95% CrI. Grey line shows the true values used for the simulation. (**D**) Shaded regions show posterior distributions of estimated antibody kinetics parameters. Dashed lines show the true value used for simulation. Note the x-axis range is small relative to the prior ranges in S5 Table. The data underlying this figure can be found at https://doi.org/10.5281/zenodo.12795911.
(TIF)

**S31 Fig. Assessment of model fitting accuracy based when fitting a model with a smoothed antigenic map to data simulated using a punctuated antigenic map.** (**A**) Model-predicted titres compared to observed HI titres at each sampling time for 3 individuals. (**B**) Posterior median and 95% credible intervals (CrI) for the cumulative number of infections over time from birth (orange). Blue solid line shows the true, known cumulative number of infections. Purple dashed line shows the time of birth. (**C**) Posterior estimated per-capita per-3-month attack rates. Red line and shaded region shows posterior median and 95% CrI. Grey line shows the true values used for the simulation. Purple dashed lines show cluster transitions. (**D**) Shaded regions show posterior distributions of estimated antibody kinetics parameters. Dashed lines show the true value used for simulation. Note the x-axis range is small relative to the prior ranges in S5 Table. The data underlying this figure can be found at https://doi.org/10.5281/zenodo.12795911.
(TIF)

**S1 Text. Description of methods to estimate strain-specific measurement offsets used in the main model.**
(DOCX)

**S2 Text. Description of full simulation-recovery analyses using the *serosolver* model.**
(DOCX)

**S1 Video. Distribution of quarterly attack rates by location over time.** Each coloured point shows the inferred attack rate in each of the 40 locations, with size and shading reflecting the posterior median attack rate. Underlying the plot is a map of the study area, with each grid cell

shaded by its log10 population density. The data underlying this figure can be found at https://doi.org/10.5281/zenodo.12795911.
(MP4)

**S1 Table. Self-reported vaccination status at time of first study visit (top) and between study visits (bottom).** Percentages exclude individuals who declined to answer, were unsure, or had missing data. The data underlying this table can be found at https://doi.org/10.5281/zenodo.12795911.
(XLSX)

**S2 Table. Estimated attack rates and infection patterns 2010–2014 prior to removing runs of consecutive infections.** Percentages shown are posterior median and 95% credible intervals. Attack rate was defined as the proportion of individuals who were infected at least once in that year. "Reinfected" gives the percentage of people that were infected more than once in a year. Bottom table shows the percentage of individuals that were infected at least once per year in 0, 1, 2, 3, 4, or 5 years between 2010 and 2014 inclusive. The data underlying this table can be found at https://doi.org/10.5281/zenodo.12795911.
(XLSX)

**S3 Table. Strain-specific measurement offset terms used in the main model fits.** Values shown are maximum posterior probability estimates from a less flexible version of the model fit to the same data (described in S1 Text). Values shown to 3 significant figures.
(XLSX)

**S4 Table. Estimated antibody kinetics parameters under the model without strain-specific measurement offsets.** The data underlying this table can be found at https://doi.org/10.5281/zenodo.12795911.
(XLSX)

**S5 Table. Description of antibody kinetics parameter values used for the simulation.** Uniform priors were used for all parameters. The data underlying this table can be found at https://doi.org/10.5281/zenodo.12795911.
(XLSX)

## Author Contributions

**Conceptualization:** James A. Hay, Kin On Kwok, Adam Kucharski, Jonathan M. Read, Justin Lessler, Derek A. T. Cummings, Steven Riley.

**Data curation:** James A. Hay, Huachen Zhu, Chao Qiang Jiang, Kin On Kwok, Ruiyin Shen, Jonathan M. Read, Justin Lessler, Derek A. T. Cummings, Steven Riley.

**Formal analysis:** James A. Hay, Huachen Zhu, Kin On Kwok, Adam Kucharski, Jonathan M. Read, Justin Lessler, Derek A. T. Cummings, Steven Riley.

**Funding acquisition:** Kin On Kwok, Jonathan M. Read, Justin Lessler, Derek A. T. Cummings, Steven Riley.

**Investigation:** James A. Hay, Huachen Zhu, Chao Qiang Jiang, Kin On Kwok, Ruiyin Shen, Adam Kucharski, Bingyi Yang, Jonathan M. Read, Justin Lessler, Derek A. T. Cummings, Steven Riley.

**Methodology:** James A. Hay, Huachen Zhu, Chao Qiang Jiang, Kin On Kwok, Adam Kucharski, Bingyi Yang, Jonathan M. Read, Justin Lessler, Derek A. T. Cummings, Steven Riley.

**Project administration:** Huachen Zhu, Chao Qiang Jiang, Kin On Kwok, Jonathan M. Read, Justin Lessler, Derek A. T. Cummings.

**Resources:** Huachen Zhu, Chao Qiang Jiang, Kin On Kwok, Jonathan M. Read.

**Software:** James A. Hay, Adam Kucharski.

**Supervision:** Chao Qiang Jiang, Steven Riley.

**Validation:** James A. Hay.

**Visualization:** James A. Hay.

**Writing – original draft:** James A. Hay, Jonathan M. Read, Steven Riley.

**Writing – review & editing:** James A. Hay, Kin On Kwok, Ruiyin Shen, Adam Kucharski, Bingyi Yang, Jonathan M. Read, Justin Lessler, Derek A. T. Cummings, Steven Riley.

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
