## [Editor Report · Decision Letter 0]

29 Mar 2024

Dear Dr Hay, 

Thank you for submitting your manuscript entitled "Reconstructed influenza A/H3N2 infection histories reveal variation in incidence and antibody dynamics over the life course" for consideration as a Research Article by PLOS Biology.

Your manuscript has now been evaluated by the PLOS Biology editorial staff, as well as by an academic editor with relevant expertise, and I'm writing to let you know that we would like to send your submission out for external peer review.

Once your full submission is complete, your paper will undergo a series of checks in preparation for peer review. After your manuscript has passed the checks it will be sent out for review. To provide the metadata for your submission, please Login to Editorial Manager (https://www.editorialmanager.com/pbiology) within two working days, i.e. by Apr 03 2024 11:59PM.

Kind regards,

Roli Roberts

Roland Roberts, PhD

Senior Editor

PLOS Biology

rroberts@plos.org

---

## [Decision Letter · Decision Letter 1]

24 May 2024

Dear Dr Hay,

Thank you for your patience while your manuscript "Reconstructed influenza A/H3N2 infection histories reveal variation in incidence and antibody dynamics over the life course" went through peer-review at PLOS Biology. Your manuscript has now been evaluated by the PLOS Biology editors, an Academic Editor with relevant expertise, and by three independent reviewers.

As you will see in the reports, all reviewers are positive about the work, but have further suggestions and concerns that should be addressed before the work can be accepted for publication. Specifically, reviewer #1 request additional information in the text and some improvements to the figures, reviewer #2 would like some clarification about the role of cross-reactivity antibodies in the model and reviewer #3 wonders about the role of antigenicity on H3N2 circulation, and if predictions can be made using the model. The Academic Editor has also provided comments, highlighting some overstatements that need nuancing or further discussion (see the end of this email, below the reviewer reports). Addressing the concerns of all reviewers, as well as those of the Academic Editor, will be required for a successful revision.

IMPORTANT: Please also address the following editorial notes:

Please supply the numerical values either in the a supplementary file or as a permanent DOI’d deposition for the following figures and tables:

Figure 1ABC, 2A, 3ABCD, 4, 5, S1, S2, S3, S4AB, S5AB, S6, S7AB, S8, S9ABC, S10, S11, S12ABCD, S13BCDEF, S16, S17AB, S18AB, S19AB, S20, S21, S22, S25ABCD, S26ABCD, S27ABC, S28ABCD, S29ABCD.

Table 1, 2, S1, S2, S4, S5.

c) Please cite the location of the data clearly in all relevant main and supplementary Figure legends, e.g. “The data underlying this Figure can be found in S1 Data” or “The data underlying this Figure can be found in https://doi.org/10.5281/zenodo.XXXXX”

d) Please ensure that your Data Statement in the submission system accurately describes where your data can be found and is in final format, as it will be published as written there.

e) We thank you for providing the code used on Github. However, please note that we cannot accept sole deposition of code in GitHub, as this could be changed after publication. However, you can archive this version of your publicly available GitHub code to Zenodo. Once you do this, it will generate a DOI number, which you will need to provide in the Data Accessibility Statement (you are welcome to also provide the GitHub access information). See the process for doing this here: https://docs.github.com/en/repositories/archiving-a-github-repository/referencing-and-citing-content

f) Please note that per journal policy, the model system/species (human) studied should be clearly stated in the abstract of your manuscript.

**IMPORTANT - SUBMITTING YOUR REVISION**

*Resubmission Checklist*

*Published Peer Review*

*PLOS Data Policy*

*Blot and Gel Data Policy*

Sincerely,

Melissa

Melissa Vázquez Hernández, PhD

Associate Editor, PLOS Biology

on behalf of

Roli

Roland Roberts, PhD

Senior Editor

PLOS Biology

rroberts@plos.org

REVIEWERS' COMMENTS:

Reviewer #1: 

In this study, the authors further developed their own software package to model antibody kinetics against antigenically different H3N2 viruses, and applied it to a large dataset including >60,000 serum hemagglutination inhibition (HI) titers from >1,000 participants around Guangzhou, China. They also compared the new dataset with their previous smaller dataset from Viet Nam. Interestingly, the authors successfully focused on the 'infection' history inference, by utilizing mostly 'vaccine-naïve' populations for the study cohort. By reconstructing life-time infection history at individual and at population levels from the serological data against a panel of historical H3N2 virus strains, the authors found that estimated attack rates could be much higher than values based on surveillance, presumably due to asymptomatic infection. Their model fits also estimated the HI titers at different timepoints for individuals, which enabled the authors to infer the time of the previous infection events and relationship between titers and infection probability. The correlations between HI titers and protection among different age groups suggests that non-HI immunity contributed differently in different age groups. The data interpretation by the authors is generally careful and thorough, with plenty of references cited. Although there are several limitations because these types of models are based on only seroconversion and simplified factors, the authors appropriately recognized and discussed such points.

Specific points:

- The details of the experimental conditions for the hemagglutinin inhibition assays should be described in the method section, not by just citing reference #41, since such information is important for readers to see if the data curation steps are scientifically appropriate.

- Figure 1: A and B, and C could be presented with different color-coding to improve reader understanding, since they are different parameters. Also, the labels on the color scale in Figure 1C could be shown as percentages, to avoid confusion.

- Figure S6: Comparison of annual attack rates between the two cohorts. Some descriptions in the figure legend are highly subjective (e.g., remarkably similar, very different, etc.). This reviewer is wondering whehter the data could be statistically analyzed.

- Figure S15: "PE06" in the figure labels should be "PE09".

Reviewer #2: 

In this manuscript by Hay et al., the authors attempt to reconstruct H3N2 infection histories using HI data from a large cohort in China. The authors analyze the relative attack rates of H3N2 and find an age-dependent risk, with more infections occurring in childhood and flattens to ~2/decade. There is an impressive amount of analyses performed to help form conclusions. Major concerns arise from basic information about study design, including samples and analysis of location, as well as concerns about the predicted infections without controlling for cross-reactive antibodies. 

Major Concerns

1. Figure 2A - Is influenza seasonal in this cohort? Relative to the Ha Nam cohort, can the differences in influenza seasonality, or lack thereof, explain similarities or differences in attack rates? Is the attack rate predicted to be higher in tropical/subtropical regions relative to regions with seasonal outbreaks?

2. The rationale for looking at location is unclear. The authors segment much their research based on location, which is really just the distance from the city center. Further clarification for these analyses is appreciated. 

3. The comment on page 15 about infection appears to happen shortly after birth and contradicts reference 43 - I do not think this conclusion can be made. It is clear that children can mount antibody responses to viruses prior to their birth, suggesting some level of cross-reactivity (figure 1). This could artificially move their predicted first infection up, when in reality it could be more in line with the finding in reference 43, which is longitudinal and looking for seroconversion in cohorts of relevant ages.

4. Unclear how the effects of crossreactive antibodies on attack rate estimates is being controlled. 

Minor Concerns

1. Figure 4, it would be helpful to put the year of birth for each donor.

2. Figure 4, the lab "probability that infection occurred" is misleading. This makes it sound certain. I would suggest "predicted probability that infection occurred." 

3. A comment on the high CI shaded areas around HI titers of 7-8 in Figure 5 would be appreciated. I assume there were limited datapoints.

4. Figure 5, log base what? 

5. Figure 5, it would also be appreciated to add the relative risk at 40 HI titer for each graph.

Reviewer #3: 

Hay and colleagues leverage serum samples collected as a part of the Fluscape cohort study, perform hemagglutination inhibition (HI) assays across 20 antigenically unique viruses (1968-2014) and integrate these data into models that infer past infection events, define past attack rates at the population level across an age stratified cohort, and predict the likelihood of infection based on HI titer. Methodologies developed, adapted, and validated herein are predicted to be useful for understanding past population-level infection histories, current epidemiological trends and past or present susceptibility to infection. As the authors note, these approaches capture data that may otherwise be lost due to surveillance efforts that have largely relied upon symptomatic illness.

The manuscript is a tome, but is well written in language that is broadly understandable. I offer these suggestions to further solidify its conclusions or impact. I thank the authors for their descriptions and explanations of their models in clear and understandable language. The careful statements of chosen parameters, their meaning, and limitations and caveats were welcome. 

Major:

The authors find a high frequency of infections within this cohort over the study window. Reinfections were also greater post 2008. This 2008-present time period corresponds to a point at which human H3N2 viruses developed a complex circulation history that involved multiple antigenically distinct clades. Given the assays and models are sensitive to antigenicity, could these observations i) be due to increased infections due to increased circulating antigenic diversity and ii) in part inflated due to the lack of representation of multiple clades (e.g. 3C.3a and its descendants) that circulated during the study window. While this may not be able to be addressed experimentally it could be discussed or modeled. 

Many of the results here align with previous observations. These observations lend support to this approach, its scalability and utility of the models employed. This work would be greatly enhanced if it had demonstrated predictive value. I do not know if the following can be addressed with current models and donor/sample information. It would be interesting if predictions could be made based on the first sample collection of the likelihood of infection before the second sample collection. Similarly, if known vaccinations during the study window could be identified. If these are not tractable, a bit of discussion of what data would need to be captured or models developed to infer population-level/age-group risk of infection. 

Minor:

Table 2: Can the authors describe how infections in 2011 and 2013 were inferred without using viruses from these years in their panel. 

Figure 2: Denote what the gray shading is for. I am assuming it corresponds to dates before a donor's birth. 

Figure S3. What accounted for the higher HAI titers for visit 2 and perhaps broader serum antibody reactivity?

ADDITIONAL COMMENTS FROM ACADEMIC EDITOR:

The authors should pay particular attention to R2's concerns about young children. The authors seem to have taken no account of maternal antibodies, and they should be more cautious about this conclusion, and discuss maternal antibodies as a limitation.

I also urge the authors to be more careful about language that is suggestive of accepting the null hypothesis (whether or not this acceptance is associated with a test): "not predicted by"; "no more variation"; "did not decline". We don't know if the estimated decline is real (nor would we be sure it didn't happen even if it were actually not observed at all in this particular study): better to say something like “with no clear association with distance”.

---

## [Decision Letter · Decision Letter 2]

26 Sep 2024

Dear Dr Hay,

Thank you for the submission of your revised Research Article "Reconstructed influenza A/H3N2 infection histories reveal variation in incidence and antibody dynamics throughout life" for publication in PLOS Biology. On behalf of my colleagues and the Academic Editor, Jonathan Dushoff, I'm pleased to say that we can in principle accept your manuscript for publication, provided you address any remaining formatting and reporting issues. These will be detailed in an email you should receive within 2-3 business days from our colleagues in the journal operations team; no action is required from you until then. Please note that we will not be able to formally accept your manuscript and schedule it for publication until you have completed any requested changes.

IMPORTANT:

a) Note that we have made a slight change to the end of your Title, as we felt that "...over the life course" read slightly oddly.

b) You hadn't supplied a blurb for us to use for social media, homepage, eToCs, etc. We fashioned the following from elements of the Abstract, but realise that it may not be capturing what *you* feel to be the most important findings. Please feel free to email me (rroberts@plos.org) with a new version, preferably slightly shorter:

"Humans experience many influenza infections over their lives, but how are the resulting within-host immunological processes linked to population-level epidemiology? This study of lifetime influenza A/H3N2 infection histories for 1,130 people allows reconstruction of historical seasonal influenza patterns, and of each individual's expected antibody profile over their lifetime."

Sincerely, 

Roli Roberts

Senior Editor

PLOS Biology

rroberts@plos.org